# Gender-Related Differences in the Citation Impact of Scientific Publications and Improving the Authors' Productivity

**Oleksandr Kuchanskyi** [1], **Yurii Andrashko** [2], **Andrii Biloshchytskyi** [3,4,*], **Serik Omirbayev** [3], **Aidos Mukhatayev** [5,6], **Svitlana Biloshchytska** [4,7] **and Adil Faizullin** [3]

1   Department of Information Systems and Technology, Taras Shevchenko National University of Kyiv, 01601 Kyiv, Ukraine; kuchansky@knu.ua
2   Department of System Analysis and Optimization Theory, Uzhhorod National University, 88000 Uzhhorod, Ukraine; yurii.andrashko@uzhnu.edu.ua
3   University Administration, Astana IT University, Astana 010000, Kazakhstan; serik.omirbayev@astanait.edu.kz (S.O.); adil.faizullin@astanait.edu.kz (A.F.)
4   Department of Information Technology, Kyiv National University of Construction and Architecture, 03680 Kyiv, Ukraine; bsvetlana2007@gmail.com
5   Center Administration, Northern Education Development National Center, Astana 010000, Kazakhstan; mukhatayev.aidos@gmail.com
6   Department of Science and Innovation, Astana IT University, Astana 010000, Kazakhstan
7   Department of Computational and Data Science, Astana IT University, Astana 010000, Kazakhstan
*   Correspondence: bao1978@gmail.com

**Abstract:** The article's purpose is an analysis of the citation impact of scientific publications by authors of different gender compositions. The page method was chosen to calculate the citation impact of scientific publications, and the obtained results allowed to estimate the impact of the scientific publications based on the number of citations. The normalized citation impact is calculated according to nine subsets of scientific publications that correspond to patterns of different gender compositions of authors. Also, these estimates were calculated for each country with which the authors of the publications are affiliated. The Citation database, Network Dataset (Ver. 13), was chosen for the scientometric analysis. The dataset includes more than 5 million scientific publications and 48 million citations. Most of the publications in the dataset are from the STEM field. The results indicate that articles with a predominantly male composition are cited more than articles with a mixed or female composition of authors in this direction. Analysis of advantages in dynamics indicates that in the last decade, in developed countries, there has been a decrease in the connection between the citation impact of scientific publications and the gender composition of their authors. However, the obtained results still confirm the presence of gender inequality in science, which may be related to socioeconomic and cultural characteristics, natural homophily, and other factors that contribute to the appearance of gender gaps. An essential consequence of overcoming these gaps, including in science, is ensuring the rights of people in all their diversity.

**Keywords:** PageRank; gender inequality; citation impact; scientific research; research productivity; scientometrics





## 1. Introduction

New knowledge, ideas, and innovations are created thanks to the development of scientific cooperation. Scientific cooperation is a joint activity of scientists to create and verify new knowledge. The results of scientific cooperation are the publication of scientific articles, the organization and implementation of joint scientific projects, and the organization of conferences, seminars, and other scientific events. The increase in the productivity of the scientific activity of individual scientists and scientific teams is a factor that affects the development of innovations in the region and the state as a whole. The current direction of scientometrics is to identify the influence of demographic, social, and gender differences

on publishing productivity. In works [1,2], it was determined that the form and intensity of scientific cooperation affect publishing productivity and the creation of innovations [3]. This process is significantly influenced by the peculiarities of the construction of the social space in which scientific teams cooperate. It can be assumed that one of the influencing factors in forming patterns of scientific collaboration is gender. The impact of gender differences on publication productivity and citation of scientific publications is described in [4]. In work [5], it was found that gender-heterogeneous working groups allow for the production of scientific results of higher quality. However, it is complicated by natural gender homophily [6]. The ability to collaborate with peers also manifests itself in citations of scientific publications. In work [7], scientists tend to cite publications by authors of the same gender as themselves. Gender-based questions about homophily in research are described in works [8,9].

Ensuring respect for human dignity, equality, and rights is a critical value of the EU and other countries with a high human development index. An essential condition for ensuring these values is the implementation of a policy of gender equality and the elimination of gender gaps. In recent decades, there has been a growing trend to reduce the influence of gender differences among researchers when forming the composition of scientific projects. In particular, work [10] indicated that the influence of gender differences on scientific publication productivity is decreasing in current conditions, especially among young scientists. The analysis in [10] claims that gender differences in the productivity of scientific activity have been disappearing recently. A few decades ago, the number of scientific publications with male authors significantly exceeded that of female authors, but now this trend has changed. However, it was difficult for women to obtain positions in science for a long time since this field was almost entirely male-dominated [11]. However, even with the gender representativeness of the STEM direction in education and science, this process was accompanied by increased gender differences in productivity and influence [12].

The prevailing situation is that there are fewer females than males in the higher ranks in academic circles. In work [13], it is indicated that, personally, females with high scientific results in a scientific group significantly influence the productivity of this group. In work [14], it is indicated that this is influenced by the higher emotional intelligence of females compared to males. Ensuring gender diversity in educational and scientific spaces is complex and multifaceted. Some aspects of gender diversity policy in university networks are described in [15]. It is important to note that gender representativeness can differ in different science areas. In work [16], a study of the results of the work of 150,000 mathematicians was conducted. It has been shown that females publish less early in their careers and drop out of research faster than males. As a result, top mathematics journals publish fewer articles authored by women. A similar trend can be observed in the direction of computer science. However, this is a separate research task.

Even though the trend of overcoming gender gaps is one of the priorities in developed countries, questions remain as to whether scientific publications with a different gender composition are cited differently. And if so, what could it be connected with? To find answers to this question, choose a method with which you can effectively evaluate citation impact. Traditionally, citation impact is defined as the number of times subsequent publications cite a publication.

One of the methods that can be used to evaluate the scientific publication productivity or citation impact of a scientist is the PageRank method [17]. The traditional purpose of the PageRank method is to determine the influence of a user on social networks or to evaluate the importance of web pages. Each network user or page is assigned an actual number that measures importance or reputation. The larger this number, the higher the importance [18]. There are modifications to the PageRank method to calculate the productivity of scientific activities, the citation index, the reputation of scientific journals, etc. The classical PageRank method uses only edge relations and does not consider higher-order structures, particularly subgraphs. One of the concepts of modifying the PageRank method, described in [19], is the complication of the evaluation calculation by including higher-order structures in

the calculation. In work [19], it is shown that this approach helps rank social network users better. This approach makes sense because citation networks tend to have a complex structure. This fact can be considered to assess the impact of citations in practice. However, it is challenging to use this method in real-time. A dynamic change in the structure of the citation network leads to the need to recalculate the scores, which is cumbersome.

In [20], an iterative method for calculating PageRank is proposed, simplifying the rating calculation. In general, using the PageRank method allows you to consider all the information about all the citations of the network authors when evaluating them. While the h-index [21] and its analogs, such as the i10-index, g-index, etc., when calculating the productivity of scientific activity, lose part of the citations outside the core, the work [22] describes the method of calculating the scientific productivity of collective subjects (universities, scientific institutes, departments, faculties, etc.) based on the Time-Weighted PageRank Method with Citation Intensity (TWPR-CI). It is shown that the advantage of the TWPR-CI method is the higher sensitivity of the scientific productivity estimates for new collective subjects that it averages during the first ten years of observation. The method's sensitivity is essential and can be used for citation impact evaluation, especially for recently published posts. However, the number of citations of new publications may be small, so this method will not differ from the classic PageRank method.

An analysis of the continuity of research in intergender scientific cooperation [23] is a direction that allows a better understanding of the features of the involvement of scientists of different genders in joint scientific projects. Well-known methods of researching patterns of scientific cooperation and choosing scientists for the organization of projects [23,24] can also be used to study the influence of gender on scientific interaction. Also, the methods described in works [25–30] can be used to evaluate the productivity of scientific activity, management, and competence selection of project executors using a gender approach. The work [12] describes a thorough study of the impact of gender inequality on scientific careers in different countries. It found that the increase in female participation in science over the past 60 years has been accompanied by a widening of the gender gap in both scientific productivity and impact. The article hypothesizes that the gender composition of authors of scientific publications has an impact on citation rates. If the influence is detected, it may mean that the gender composition of scientific teams working on joint research affects their scientific publication productivity. This trend may differ depending on the countries and areas of scientific research and may change over time. Accordingly, the article's goal is an analysis of the citation impact of scientific publications by authors with different gender compositions. Also, this article does not suggest that biases are conscious and that biases may depend on other socioeconomic and cultural factors but allow for the reveal of existing inequalities. Identified differences in the citation of scientific publications are not a sign of discrimination based on gender but are an indicator that captures the current state of publication activity.

A citation dataset of scientific publications was investigated. The Network Dataset (13 versions) consists of more than 5 million scientific publications and 48 million citations [31], collected from databases such as DBLP [32], ACM [33], Microsoft Academic Graph [34], and others. The construction of the database is described in more detail in [35]. The following research stages were implemented:

1. Calculate the citation impact for each scientific publication in the citation network. To calculate citation impact, the number of citations of scientific publications was counted. Also, for citation impact calculation, the PageRank method was used [36,37].
2. All publications are divided into eight classes according to the gender composition of the authors of these publications. The publication belonging to the corresponding cluster is determined by the author's article based on a unique service for determining the gender of a person by their first name.
3. To examine the dependency of scientific publications' citations based on the gender composition of their authors, the obtained results for eight classes are compared among each other. Special attention should also be paid to citation scores' impact on scientific

publications by authors from different countries. Analyzing the change in citation scores' impact on scientific publications from different countries is also essential.

Researching the influence of gender differences on scientific publication productivity is relevant for the development of innovations and scientific production in general. The identified gender inequality in the academic circle should be eliminated at the institution of higher education or scientific research institution and at the state level. An increase in the scientific publishing activity of the authors contributes to the growth of the scientific productivity of the institutions with which these authors are affiliated. The described study continues the research published in previous works [22,37].

## 2. Methods and Data

### 2.1. Basic Terms and Concepts

Some terms and concepts have been used in the publication. Citation impact is determined by the number of times subsequent publications cite a publication. This study used the PageRank method to calculate the citation impact of scientific publications. The citation impact of a scientific publication, which was calculated as a result, is called PageRank citation impact. Also, the traditional method of calculating their total number of citations was used to evaluate the impact of scientific publications.

The work focuses on the calculation of the citation impact of scientific publications with different gender compositions. It is important to understand the regional distribution by country and the change over time in the intensity of citation of scientific publications with different gender compositions: male, female, and mixed.

Patterns for the gender composition of authors were highlighted. Each pattern corresponds to a specific class in which scientific publications were included. Each of these classes is studied separately. The evaluation of the citation impact of scientific publications by authors from different countries was conducted using open data collected over a long period. This allows you to investigate the change of citation impact of scientific publications for different classes over time. Also, sufficient data allow us to analyze the citations separately and the impact of scientific publications in different countries.

The work examines eight patterns for the gender composition of authors of scientific publications. It is assumed that a particular pattern will determine each article, and the citation score impact for these articles will differ. All scientific publications are divided into eight classes or subsets for each pattern separately. Let $S = \{s_1, s_2, \ldots, s_n\}$ be the set of scientists, and n is the number of scientists. Let $P = \{p_1, p_2, \ldots, p_m\}$ be the set of scientific publications published by scientists from set S, and let m be the number of scientific publications. With each publication $p_j, j = \overline{1, m}$ one or more authors of this publication are associated. We set the function $F \subseteq S \times P$, which the set of pairs will determine $\left(s_i, p_j\right)$, $i = \overline{1, n}, j = \overline{1, m}$. Let us set the function: $g : S \to \{f, m\}$ determines the gender of each scientist from the set S. Then, define a tuple: $\Delta\left(p_j\right) = \left\langle g(s_i) \middle| \left(s_i, p_j\right) \in F, i = \overline{1, n}, j = \overline{1, m} \right\rangle$.

If for scientific publications $p_k, k = \overline{1, m}, p_k \in P, \forall d \in \Delta(p_k), d = f, \mathrm{card}(\Delta(p_k)) > 1$, then all authors of scientific publications $p_k$ are women and publications belong to the pattern "Fff". If $\mathrm{card}(\Delta(p_k)) = 1$ then publications belong to the pattern "F". If $\forall d \in \Delta(p_k)$, $d = m, \mathrm{card}(\Delta(p_k)) > 1$, then the authors of the scientific publications $p_k$ are male, and, accordingly, the publications belongs to the "Mmm" pattern. If $\mathrm{card}(\Delta(p_k)) = 1$, the publication belongs to the "M" pattern. Other patterns are described in Figure 1. A capital letter at the beginning of the pattern's name indicates the gender of the first author of the scientific publication, respectively, F—female, M—male. The analysis of the specified number of classes or subsets of scientific publications corresponding to the specified patterns is sufficient for the study.

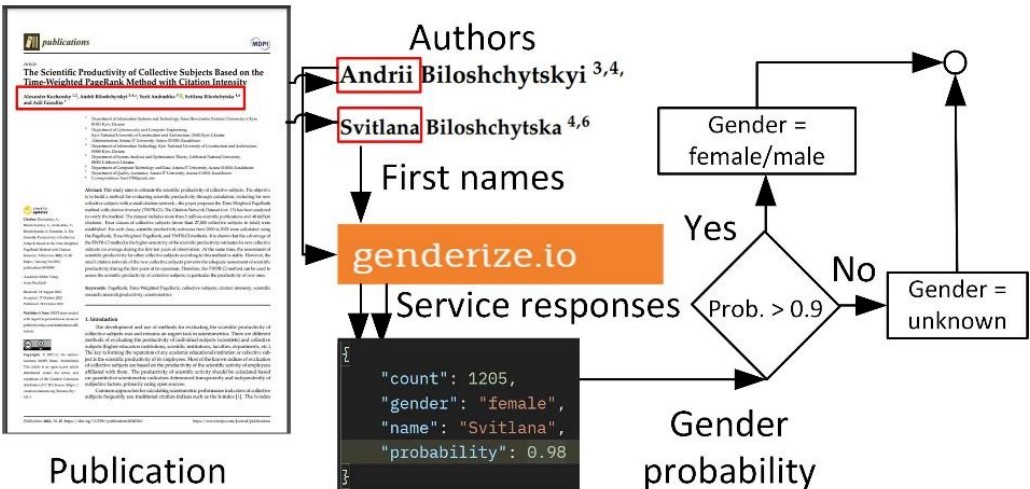

**Figure 1.** Conceptual diagram of the method of determining the gender composition of authors of scientific publications [22].

It should be noted that the gender composition of publications is determined based on a service that checks the gender of the authors of these publications. Separately, a significant number of publications with an uncertain gender composition should be considered when at least for one author, the service cannot identify the author's gender with sufficient accuracy. It should also be understood that the obtained results may have some deviations since, among the authors, a certain number of persons may identify themselves as non-binary. Still, the first name cannot determine it.

## 2.2. The Assessment of Citation Impact and PageRank Citation Impact of Scientific Publications

To calculate the citation impact for each scientific publication, you need to calculate the number of citations of this publication in other scientific publications. This indicator shows the influence of a scientific publication. The higher the citation impact of a scientific publication, the greater the influence of this publication. If $Q^{CI} = \{q_1, q_2, \ldots, q_m\}$ is the citation scores impact for each scientific publication $p_j$, $j = \overline{1, m}$, $Q^{CI} : P \to \mathbb{N} \cup \{0\}$. This indicator only shows the total number of citations, and it can quantify this publication's interest among other relevant authors.

The PageRank method was used to evaluate the influence of scientific publications. This method allows you to determine the impact of a scientific publication in comparison with other publications under consideration. According to the PageRank method, the scalar evaluation of the citation impact of a scientific publication $p_j$ is $j = \overline{1, m}$ calculated according to the formula:

$$r_j = \sum_{y=1}^{m} \beta_{jy} \xi_y r_y, \ j = \overline{1, m}, \tag{1}$$

where $r_j$ is the PageRank score citation impact of a scientific publication $p_j$, $j = \overline{1, m}$, $\beta_{jy}$, $j = \overline{1, m}$, and $y = \overline{1, m}$ are the coefficient that determines the presence of a scientific publication, $p_j$, $j = \overline{1, m}$ is the list of publication citations $p_y$, and $y = \overline{1, m}$, $\xi_y$ is a coefficient that ensures the existence of a non-trivial solution of the system of linear algebraic Equation (1).

As a result of applying Formula (1), a homogeneous system of linear algebraic equations is constructed:

$$Br = 0, \tag{2}$$

where B is the matrix of coefficients of the system of the form:

$$B = E - \left\{ \beta_{jy} \xi_y \right\}_{j,y=1}^{m},$$

where E is the single matrix, and $r = w^T$ is a column vector unknown of grades, $w = (r_1, r_2, \ldots, r_m)$.

For there to be a non-trivial solution of the system of algebraic Equation (1), the matrix B must be degenerate, i.e., $\det(B) = 0$.

Let us ask a subset of the Cartesian product $C \subset P \times P$, which determines the citation of publications $P \times P = \left\{ \left( p_j, p_y \right) \middle| p_j, p_y \in P, j \neq y \right\}$. From plural scientific publications which are cited by a given publication $p_j \in P$, we define through $C\left( p_j \right) = \left\{ p_y \in P \middle| \left( p_j, p_y \right) \in C, y = \overline{1, m} \right\}$. The formulas can determine the coefficients of system (1):

$$\beta_{jy} = \begin{cases} 1, \text{ if } p_j \in C\left( p_y \right) \\ 0, \text{ if } p_j \notin C\left( p_y \right) \end{cases}, \tag{3}$$

$$\xi_y = \left\| C\left( p_y \right) \right\|^{-1}, \ y = \overline{1, m}, \tag{4}$$

where $\beta_{jy}$ is the indicator of the presence of the publication $p_j$ in the list of publication references $p_y$, and $\xi_y$ is the value inverse of the total number of citations in the publication $p_y$.

After finding the estimates, it is advisable to standardize them according to the formula

$$r'(p_i) = r_i \left( \sum_{j=1}^{m} r_j \right)^{-1}, \ i = \overline{1, m}, \tag{5}$$

where $r_i$ is the PageRank score citation impact of a scientific publication $p_i$, and $i = \overline{1, m}$, $r'(p_i)$ is the normalized PageRank score citation impact of a scientific publication $p_i, i = \overline{1, m}$.

The more citations a scientific publication has over time, the higher its citation impact. Therefore, to evaluate the citation impact of a scientific publication, you can count the number of citations of this publication. The advantage of calculating the citation impact of a scientific publication using the PageRank method is that this method considers the influence of a scientific publication by the number of citations compared with the citations of other scientific publications.

The citation base of scientific publications was analyzed in the Network Dataset (ver. 13), and a citation network was built. Next, the citation score was calculated for all scientific publications based on the number of citations and PageRank rating citation impact of all scientific publications. It is necessary to solve the system of linear algebraic equations of large dimensions (2) to find the PageRank score citation impact. The iterative process of the Gauss–Seidel method is used to find the approximate solution of the system of linear algebraic Equation (2). At step zero, the value of the PageRank scores citation impact of all scientific publications is equal to 1. At the k-th step, the value of each PageRank score citation impact is calculated. The following formula is used to find the index of the publication:

$$r_j^k = \sum_{y=1}^{m} \beta_{jy} \xi_y r_y^{k-1}, \ j = \overline{1, m}, \ k \in \mathbb{N}, \tag{6}$$

where $r_j^k$ is the approximate value of PageRank citation impact publications $p_j$ at the k-th step, $r_j^{k-1}$ is the approximate value of the PageRank estimate citation impact publications $p_j$ at the $(k - 1)$-th step, and the coefficients are calculated according to Formulas (3) and (4).

After each step, starting from zero, the maximum relative change in citation scores was calculated to impact scientific publication according to the formula:

$$\Delta^k = \max_{j=\overline{1,m}} \left| r_j^k - r_j^{k-1} \right|, \tag{7}$$

where $\Delta^k$ is the maximum relative change in PageRank scores citation impact scientific publication $p_j$. The iterative method stops if $\exists \ \varepsilon > 0$ the maximum relative change in

citation scores impacts scientific publication $\Delta^k < \varepsilon$. The value $\varepsilon > 0$ is some small number that is specified in advance. After that, the values are normalized according to Formula (5).

A method for determining the gender composition of authors of scientific publications is proposed. The conceptual diagram of the method is shown in Figure 1. The method consists of three stages.

At the preparatory stage, PageRank scores are calculated for each scientific publication's citation impact and the citation impact calculated by the number of citations.

In the first stage, the gender identity of the authors is determined by their names using the genderize.io service [38]. This service allows you to determine with the specified accuracy whether the entered first name belongs to a male or female. First is used to determine the gender name of each author. If the name belongs to a male's name according to the genderize.io service (identification accuracy threshold exceeds 0.9), then the author is identified as a man. If the name belongs to a female, according to the genderize.io service (identification accuracy threshold exceeds 0.9), the corresponding author is identified as a female. If the identification accuracy threshold is less than 0.9, then we believe the author's gender cannot be determined. The threshold is chosen empirically since the gender of the author should be identified as accurately as possible. As already indicated, among the authors of publications, there may be a small part of those who, according to the genderize.io service, are identified as male or female, but they are not binary. Determining this fact by the first name is impossible.

In the second stage, the set of scientific publications with the known gender of the authors is divided into eight subsets (Table 1). If the gender of at least one of the authors could not be determined, then the article belongs to the subset with an uncertain gender composition of authors. Each author of a scientific publication has a specific affiliation. Accordingly, the publication belongs to those countries whose authors are affiliated with institutions of higher education or scientific institutions of these countries.

**Table 1.** Patterns of scientific publications by the gender composition of their authors.

| Pattern | Interpretation |
| --- | --- |
| Fff | all authors of a scientific publication are female (more than one author) |
| Mmm | all authors of a scientific publication are male (more than one author) |
| Fmm | all authors of the scientific publication are male except for the first author, who is female |
| Mff | all authors of the scientific publication are female except for the first author, who is male |
| Ffm | authors of scientific publications, both male and female. The first author is female |
| Mfm | the authors of the scientific publication are both male and female. The first author is male |
| F | the scientific publication has one female author |
| M | the scientific publication has one male author |

From the database of scientific publications, Citation Network, the dataset was selected from those scientific publications affiliated with the list of countries with different gender parity scores according to the Global Gender Gap Report 2022 [39]. It is necessary to check whether there is a correlation between citation scores impact of scientific publications by authors from certain countries on their gender parity score, according to the Global Gender Gap Report 2022.

Also, to establish the dynamics of changes in the citation rating impact of scientific publications of different countries over time, their evaluations were calculated for two patterns with purely male and female authors.

Jupiter notebook environment was used for scientometric analysis and dataset processing in Python programming language.

*2.3. Collection of Data*

The database of Citation publications was used for the scientometric analysis of the Network Dataset (ver. 13) of 5,354,309 scientific publications and 48,227,950 citations [31], collected from databases DBLP [32], ACM [33], Microsoft Academic Graph [34], and others. The specified version contains current data on publication citations as of May 2021.

The research used data that other researchers partially pre-processed. In particular, the considered dataset does not contain duplicate publications. Unique identifiers are assigned to each researcher and each publication. Also, only the authors' full names and their countries of affiliation were used in the study. The probability of spelling errors in these data is minimal. We also manually checked randomly selected data samples.

When determining the gender of the author, we avoided controversial points. If the genderize.io service did not indicate the gender with sufficient probability, we marked the gender of the author as unknown.

The patterns of the gender composition of the authors of these publications are defined in Table 1, and services for identifying male and female first names were used. The genderize.io service was used to compile lists of male and female first names. The genderize.io service contains data on the potential gender of 114,541,298 first names from 242 countries worldwide. Among the authors of publications in Citation Network, 451,052 unique first names were identified in the dataset, for which the gender affiliation of the authors was determined using the genderize.io service. As a result, it was established that among the authors of publications, there are 86,792 female names, 193,747 male names, and 170,513 names, the gender of which could not be established with a reliability of more than 90%. As a result of applying this method, the gender identity of all authors was established for 76.6% of publications in the selected dataset. For 23.4% of publications, it was not possible to establish gender affiliation for at least one of the authors.

To determine the gender of the authors, the use of the Gender API [40] service, which contains data on 6,084,389 first names from 191 countries, was also considered, but this service offers only 100 requests per month for free use. Therefore, it was selected for control. Among 280,539 first names of scientific publications, for which the gender of the authors was determined using the genderize.io service, 100 were randomly selected, for which the gender of the authors was determined using the Gender API service. In all 100 cases, gender identity coincided, which makes it possible to assert the sufficient reliability of the proposed method.

The space character separates author's full name into words to select the first author's name. Next, a search is conducted for each word in the list of names without considering the case of the letters. If the author's first name is not in the list of names according to the genderize.io service or only the initials are indicated, then it is considered that the gender of the author could not be established. In addition to the subsets specified in Table 1, one more subset must be constructed. This subset will include the remaining scientific publications and the gender of the authors, which could not be established by the specified method (NA).

His affiliation was determined to establish the author's affiliation with a specific country. A publication belongs to a subset of publications from a particular country if at least one of the authors is affiliated with a higher education institution belonging to that country.

## 3. Results

*3.1. Dataset Features Research*

For scientometric analysis, the entire database analyzed scientific publications in English from 1815 to 2021; however, publications and bases were unevenly distributed over time. About 90% are scientific publications published from 1998 to 2021. The quantity of publications in the Citation Network Dataset (ver. 13) by decades is shown in Figure 2.

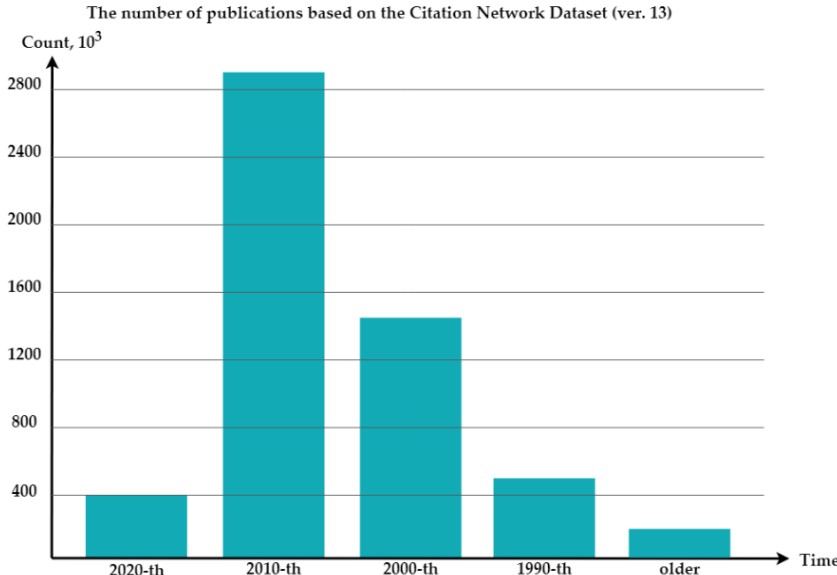

**Figure 2.** Number publications by decade based on Citation Network Dataset.

The subject areas of the publications in this database were studied separately. The central part of publications belongs to such subject areas as computer science, artificial intelligence and artificial neural networks, mathematics and discrete mathematics, optimization and combinatorics, and software engineering. The cloud of subject directions is shown in Figure 3. This study analyzed the data comprehensively, and the distribution was not carried out separately according to these directions. For visualization, data by subject were selected, including more than 200,000 publications. Relevance to the subject area was determined by the FOS parameter from the Citation database Network Dataset (Table 2). It should be noted that a scientific publication can belong to several subject areas simultaneously.

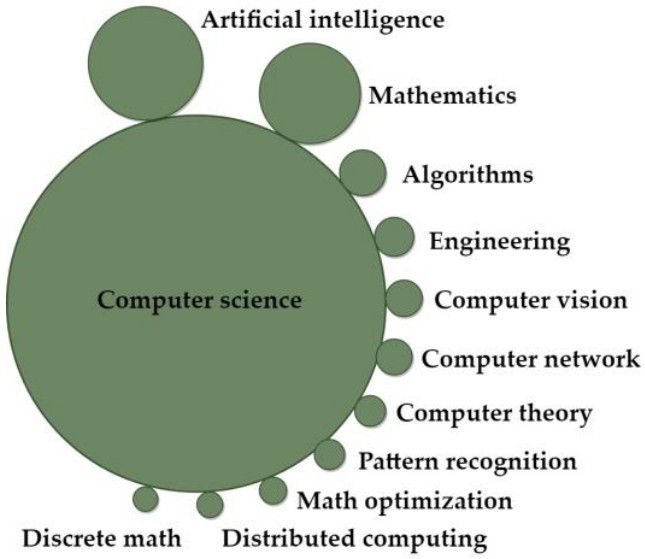

**Figure 3.** Distribution publications by subject area for Citation Network Dataset database.

**Table 2.** Number of scientific publications by different subject areas, according to the Citation database Network Dataset (displayed data by subject area with more than 200,000 publications).

| Subject Area | Count |
|---|---|
| Computer science | 3,152,625 |
| Artificial intelligence | 953,033 |
| Mathematics | 845,068 |
| Algorithm | 387,218 |
| Engineering | 325,129 |
| Computer vision | 306,614 |
| Computer network | 300,346 |
| Control theory | 259,662 |

It can be assumed that, depending on the subject area to which scientific publications belong, the gender composition of the authors of these publications may differ. In addition, citing such publications from various subject areas may have certain features. However, this is a separate research task requiring more data from other subjects.

The subject area in this dataset was already defined by the authors of the study published in [35]. Some of the specified subject areas may be part of other, more general subject areas. For example, artificial intelligence can be a subfield of computer science.

*3.2. The Results of the Calculation of PageRank Citation Impact Index and Citation Impact Index by the Number of Citations*

The Citation database Network Dataset was calculated by its citation impact according to the PageRank method and taking into account the number of citations. The accuracy of the iterative PageRank method has been established in citation impact $\varepsilon = 10^{-4}$. The maximum relative change in PageRank citation impact of a scientific publication is considered the upper estimate of the absolute error of the method. After performing six iterations of calculating the impact rating of publications by Equation (7)., the absolute error was $\Delta^6 = 2.48 \times 10^{-5}$. The authors consider this estimation accuracy sufficient, so the calculation process was completed $\Delta^6 < \varepsilon$. A citation score was also calculated to assess the impact of scientific publications based on their citations in other publications. According to this method, all scientific publications in the database are reviewed, and the number of citations of one publication in others is recorded. This number will determine the citation impact of a scientific publication.

After calculating the citation scores and impact of scientific publications among all publications in the dataset, data on publications from countries for which the research hypothesis is tested were filtered. Next, the gender identity of the authors of these publications was determined using the genderize.io service. As a result of the research, the gender identity of all authors was established for 76.6% of publications. For 23.4% of publications, it was not possible to establish gender affiliation for at least one of the authors. For each country, publications were divided into subsets according to the patterns described in Table 2. Table 3 shows the number of scientific publications whose authors are affiliated with the specified 12 countries. Data for all countries are given in Appendix A. According to the Citation database, two countries with a small number of scientific publications were included in this table Network Dataset for comparison with other countries with a significantly higher number of publications.

Statistical characteristics were calculated for the PageRank score citation impact of the scientific publications: Range (R), Mean (M), Varience (V), the number of publications out of 3σ (the number of outliers, NO), and the mean without outliers (MwO).

As a result of the calculations, it was established that there is a small number of emissions in comparison with the number of publications (NP) for each pattern. The value of the mean without outliers is less than the mean with outliers, but the ratio of calculated values between different patterns is preserved.

**Table 3.** Descriptive statistics for the PageRank score citation impact of the scientific publications.

| Pattern | NP | R | M | V | NO | MwO |
|---------|-----|-----|-----|-----|-----|-----|
| F | 121,655 | 1060.68 | 0.51239 | 19.66140 | 48 | 0.45829 |
| M | 623,914 | 1599.82 | 0.71519 | 32.40197 | 202 | 0.64416 |
| FFF | 79,694 | 282.52 | 0.50399 | 5.80887 | 141 | 0.44199 |
| MMM | 1,782,313 | 1646.24 | 0.83707 | 25.86987 | 686 | 0.77319 |
| FMM | 399,395 | 1041.94 | 0.63510 | 13.87235 | 228 | 0.58348 |
| MFF | 179,455 | 473.52 | 0.66383 | 10.79975 | 187 | 0.59149 |
| FFM | 234,434 | 214.54 | 0.48500 | 3.17360 | 901 | 0.41196 |
| MFM | 645,495 | 826.30 | 0.64434 | 11.28439 | 541 | 0.58228 |

Also, the dataset was examined to fulfill the diverse requirements within the proposed subsets defined by the defined patterns. For this, the normalized Shannon entropy was calculated using the formula:

$$H = -\frac{1}{\log_2 W} \sum_{v=1}^{W} \frac{m_v}{m} \log_2 \frac{m_v}{m},$$

where H is the normalized Shannon entropy, $m_v$ is the power of the subsets of scientific publications according to the patterns in Table 2 and the subset for which it was impossible to determine the gender composition of the authors of the publications (N/A), $v = \overline{1,W}$, $W = 9$, and m is the total number of publications. It is established that for Citation Network Data (ver. 13), H = 0.7197. This such indicator indicates sufficient representativeness of the sample to measure the representativeness with the overall population's distribution out of the scope of this research. Gender composition of authors of scientific publications by specified countries is given in Table 4.

**Table 4.** Gender composition of authors of scientific publications by specified countries.

| No | Country | All | Fff | Mff | Fmm | Mmm | Ffm | Mfm | F | M |
|----|---------|-----|-----|-----|-----|-----|-----|-----|-----|-----|
| 1 | USA | 442,281 | 7430 | 17,259 | 33,253 | 156,798 | 19,740 | 54,625 | 9153 | 45,685 |
| 2 | China | 412,520 | 5542 | 13,062 | 32,203 | 80,288 | 30,370 | 75,899 | 3298 | 9127 |
| 3 | Germany | 162,127 | 1167 | 4019 | 10,598 | 72,292 | 5175 | 18,475 | 3467 | 27,713 |
| 4 | France | 123,725 | 1633 | 4106 | 9972 | 42,829 | 6126 | 17,662 | 4410 | 18,075 |
| 5 | Japan | 110,524 | 412 | 1940 | 7775 | 59,387 | 2189 | 11,719 | 792 | 10,749 |
| 6 | G. Britain | 103,727 | 1311 | 3413 | 7782 | 34,887 | 4104 | 11,192 | 3186 | 15,937 |
| 7 | Italy | 98,243 | 2473 | 4456 | 9336 | 33,108 | 8485 | 19,035 | 1740 | 5824 |
| 8 | India | 96,816 | 2394 | 3830 | 8103 | 27,083 | 3443 | 8251 | 1007 | 4024 |
| 9 | Canada | 94,056 | 1546 | 3982 | 8290 | 36,520 | 3670 | 10,620 | 1547 | 7974 |
| 10 | Spain | 81,132 | 1157 | 2553 | 6638 | 29,979 | 5373 | 15,076 | 567 | 2824 |
| 11 | Ukraine | 1988 | 46 | 91 | 104 | 509 | 144 | 369 | 29 | 245 |
| 12 | Kazakhstan | 952 | 12 | 12 | 32 | 118 | 24 | 90 | 11 | 84 |

It is observed that for most countries, the subsets determined by patterns Mmm and M should include more publications than pattern subsets Fff and F. The requirements of the project, according to which the study was carried out, required the inclusion of research information on the countries of Kazakhstan and Ukraine. The selection of articles for Kazakhstan and Ukraine is not representative, but the general trend regarding the gender composition of the authors of the publications is visible. For each subset that corresponds to the relevant patterns of gender composition and the subset with an uncertain gender composition of authors and selected countries, the impact of scientific publications was calculated by the PageRank method and by the number of citations. Normalized citation scores' impact is given in Tables 5 and 6.

**Table 5.** Normalized relative citation scores impact of scientific publications, determined by the number of citations.

| No | Country | N/A | Fff | Mff | Fmm | Mmm | Ffm | Mfm | F | M |
|----|---------|-----|-----|-----|-----|-----|-----|-----|---|---|
| 1 | USA | 1.000 | 0.542 | 0.729 | 0.778 | 0.946 | 0.662 | 0.871 | 0.386 | 0.647 |
| 2 | China | 1.000 | 0.620 | 0.710 | 0.759 | 0.993 | 0.641 | 0.832 | 0.613 | 0.709 |
| 3 | Germany | 1.000 | 0.453 | 0.536 | 0.653 | 0.842 | 0.653 | 0.846 | 0.267 | 0.352 |
| 4 | France | 0.892 | 0.428 | 0.887 | 0.676 | 0.902 | 0.637 | 0.741 | 0.252 | 1.000 |
| 5 | Japan | 1.000 | 0.550 | 0.652 | 0.617 | 0.766 | 0.507 | 0.772 | 0.618 | 0.642 |
| 6 | G. Britain | 0.911 | 0.878 | 0.801 | 0.808 | 1.000 | 0.828 | 0.933 | 0.378 | 0.525 |
| 7 | Italy | 1.000 | 0.565 | 0.700 | 0.672 | 0.904 | 0.596 | 0.698 | 0.511 | 0.785 |
| 8 | India | 0.875 | 0.514 | 0.756 | 0.596 | 0.897 | 0.712 | 0.861 | 0.668 | 1.000 |
| 9 | Canada | 1.000 | 0.757 | 0.677 | 0.671 | 0.901 | 0.709 | 0.863 | 0.577 | 0.874 |
| 10 | Spain | 1.000 | 0.643 | 0.909 | 0.817 | 0.934 | 0.757 | 0.841 | 0.436 | 0.713 |
| 11 | Ukraine | 0.996 | 0.312 | 0.349 | 0.650 | 1.000 | 0.257 | 0.832 | 0.474 | 0.490 |
| 12 | Kazakhstan | 0.297 | 0.072 | 0.011 | 0.427 | 1.000 | 0.293 | 0.143 | 0.569 | 0.215 |

**Table 6.** Normalized relative PageRank scores citation impact of scientific publications.

| No | Country | N/A | Fff | Mff | Fmm | Mmm | Ffm | Mfm | F | M |
|----|---------|-----|-----|-----|-----|-----|-----|-----|---|---|
| 1 | USA | 1.000 | 0.515 | 0.714 | 0.732 | 0.881 | 0.562 | 0.727 | 0.460 | 0.775 |
| 2 | China | 0.989 | 0.676 | 0.786 | 0.809 | 1.000 | 0.669 | 0.829 | 0.758 | 0.970 |
| 3 | Germany | 1.000 | 0.462 | 0.528 | 0.570 | 0.722 | 0.510 | 0.649 | 0.301 | 0.377 |
| 4 | France | 1.000 | 0.876 | 0.892 | 0.694 | 0.923 | 0.600 | 0.737 | 0.299 | 0.497 |
| 5 | Japan | 1.000 | 0.602 | 0.757 | 0.629 | 0.745 | 0.482 | 0.702 | 0.709 | 0.764 |
| 6 | G. Britain | 1.000 | 0.957 | 0.836 | 0.861 | 0.997 | 0.818 | 0.878 | 0.476 | 0.582 |
| 7 | Italy | 1.000 | 0.531 | 0.646 | 0.638 | 0.821 | 0.546 | 0.642 | 0.513 | 0.775 |
| 8 | India | 0.882 | 0.568 | 0.739 | 0.595 | 0.835 | 0.639 | 0.778 | 0.736 | 1.000 |
| 9 | Canada | 1.000 | 0.711 | 0.668 | 0.642 | 0.836 | 0.605 | 0.719 | 0.578 | 0.939 |
| 10 | Spain | 1.000 | 0.672 | 0.874 | 0.784 | 0.931 | 0.675 | 0.832 | 0.603 | 0.771 |
| 11 | Ukraine | 0.996 | 0.621 | 0.324 | 0.468 | 1.000 | 0.818 | 0.654 | 0.640 | 0.533 |
| 12 | Kazakhstan | 0.297 | 0.589 | 0.000 | 0.722 | 1.000 | 0.231 | 0.256 | 0.249 | 0.497 |

The results of a pairwise comparison of publications from the represented countries from different subsets according to different patterns, on average, indicate that scientific publications with the first author, who is male or with a predominantly male composition of authors, have higher citation impact compared to publications whose authors are primarily female (Table 7). The specified trend is preserved for citation impact estimates, calculated by the number of citations and citation impact by the PageRank method. A feature has been established that the maximum number of citations of scientific publications by subset with the pattern Mmm is higher than that of scientific publications from subsets with other patterns of the gender composition of authors for most of the indicated countries. A negative value in Table 7 indicates that the specified advantage of the estimates of the two subsets is reversed. If the value of preferences in Table 7 is closer to zero, there is a bias in the citation estimates and no impact. Accordingly, scientific publications with a male and female gender composition are mainly evaluated equally.

The change in relative PageRank scores was calculated for citation impact for the period up to 2010 and from 2010 to 2021 to understand how the specified preferences change over time. The value of the benefits was determined as the difference between the average normalized ratings of the respective patterns divided by the maximum of the values. The trend of rating changes was also considered, and PageRank citation impact was determined according to different patterns. Figure 4 shows the trends of changes in the values of the evaluations of advantages $F \prec M$, $Fff \prec Mmm$ for different countries comprehensively by publications from four subsets, which patterns F, M, Fff, and Mmm determine. Such subsets of scientific publications were explicitly selected to highlight scientific publications with a purely male or female composition of authors. For subsets

Ffm $\prec$ Mfm, Fmm $\prec$ Mff, Fmm $\prec$ Mmm, and Fff $\prec$ Mff which can be seen from Table 7, preferences vary in different countries, and this change is also traced over different periods.

**Table 7.** Pairwise comparison of relative PageRank scores citation impact of scientific publications from different subsets according to defined patterns.

| No | Country | F$\prec$M | Ffm$\prec$Mfm | Fmm$\prec$Mff | Fff$\prec$Mmm | Fmm$\prec$Mmm | Fff$\prec$Mff |
|---|---|---|---|---|---|---|---|
| 1 | USA | 0.40763 | 0.22861 | −0.02635 | 0.41552 | 0.16450 | 0.28151 |
| 2 | China | 0.21950 | 0.19224 | −0.02604 | 0.31939 | 0.18918 | 0.13815 |
| 3 | Germany | 0.20118 | 0.21281 | −0.07734 | 0.35907 | 0.20930 | 0.12146 |
| 4 | France | 0.39738 | 0.19106 | 0.21598 | 0.05431 | 0.24689 | 0.01551 |
| 5 | Japan | 0.07196 | 0.30275 | 0.16964 | 0.19786 | 0.15292 | 0.21370 |
| 6 | G. Britain | 0.16081 | 0.06608 | −0.02529 | 0.04701 | 0.14478 | −0.12528 |
| 7 | Italy | 0.33818 | 0.14979 | 0.02018 | 0.35411 | 0.22785 | 0.18040 |
| 8 | India | 0.26352 | 0.17770 | 0.19945 | 0.31750 | 0.28773 | 0.23291 |
| 9 | Canada | 0.38451 | 0.15400 | 0.04888 | 0.15055 | 0.23812 | −0.05700 |
| 10 | Spain | 0.21723 | 0.18787 | 0.10011 | 0.27887 | 0.15471 | 0.23229 |
| 11 | Ukraine | 0.07203 | 0.38946 | −0.30040 | 0.15423 | −0.31861 | 0.17625 |
| 12 | Kazakhstan | 0.74135 | −0.00141 | - | 0.73906 | 0.42871 | 0.31891 |

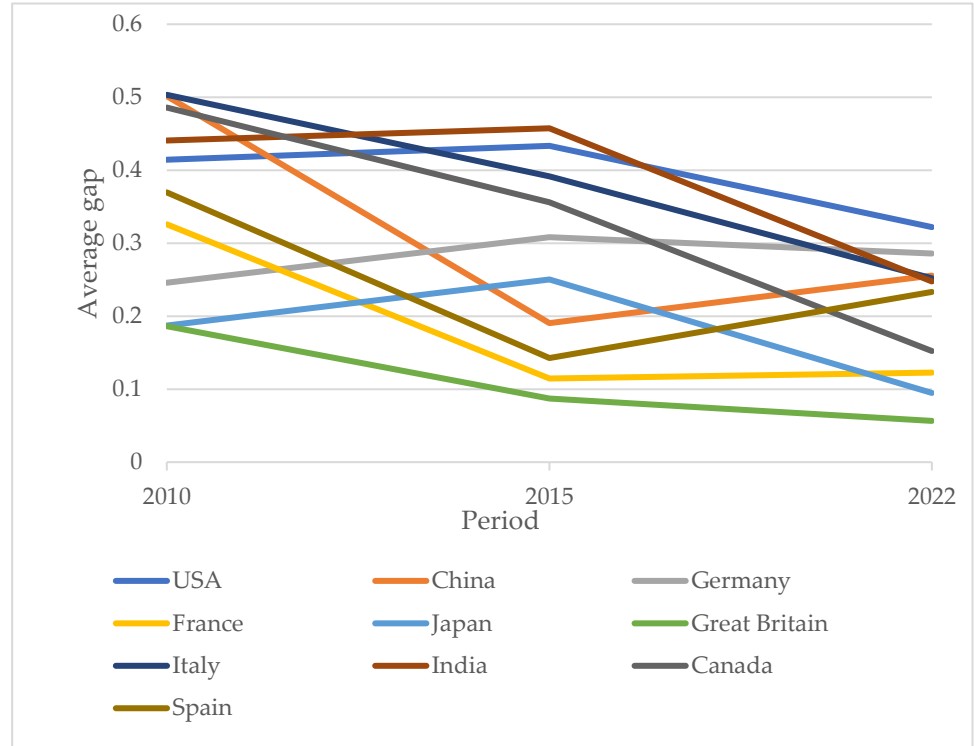

**Figure 4.** Change in the values of the preference estimates F $\prec$ M and Fff $\prec$ Mmm for different countries.

Table 8 shows the pairwise comparison of relative PageRank scores citation impact of scientific publications from different research areas according to defined patterns. The scores in the table are indicated for the areas represented by the most significant number of publications in the dataset. The research hypothesis is confirmed for all the indicated directions.

Such results can be connected to many socioeconomic factors, such as female representation in science, cultural characteristics, etc. As can be seen from Figure 4, over the last decade, the citation rate impact for scientific publications with a purely male composition of authors decreased compared to the citation impact of publications with a purely female composition of authors. In most countries in the last decade, there has been an increase in the influence of women in science and the representation of women in scientific research,

which is published in the best scientific journals. However, the state of equilibrium, i.e., the approach of preference estimates to zero, has yet to be reached for any country.

**Table 8.** Pairwise comparison of relative PageRank scores citation impact of scientific publications from different research areas according to defined patterns.

| No | Research Areas | F≺M | Ffm≺Mfm | Fmm≺Mff | Fff≺Mmm |
|----|----------------|------|---------|---------|---------|
| 1 | Computer science | 0.27503 | 0.24160 | 0.05931 | 0.37144 |
| 2 | Artificial intelligence | 0.24541 | 0.28160 | 0.09138 | 0.41557 |
| 3 | Mathematics | 0.17782 | 0.27730 | 0.03054 | 0.47048 |
| 4 | Algorithm | 0.31373 | 0.29600 | 0.15378 | 0.51274 |
| 5 | Engineering | 0.29077 | 0.21000 | 0.03516 | 0.33240 |
| 6 | Computer vision | 0.35407 | 0.26500 | 0.15399 | 0.38114 |
| 7 | Computer network | 0.22579 | 0.20350 | 0.14125 | 0.33291 |
| 8 | Control theory | 0.09866 | 0.25570 | 0.10023 | 0.26358 |
| 9 | Pattern recognition | 0.48960 | 0.35490 | 0.13328 | 0.51652 |
| 10 | Mathematical optimization | 0.34463 | 0.25860 | 0.14027 | 0.49677 |

Estimates of the preferences of subsets with different patterns by calculated citation impact can determine the availability of opportunities for females and males to participate in scientific projects and publish high-quality scientific articles. It can be assumed that in developed countries, for specific estimates of benefits F ≺ M and Fff ≺ Mmm, the value will be close to zero. This means that publications with a female and male composition are cited equally. Accordingly, the representation of females and males in science is equally high.

## 4. Discussion

### 4.1. Findings

The estimates of citation impact may, to some extent, reflect the productivity of the authors of these publications. The more the author's publications are cited, the more the author is published in the best scientific journals. Accordingly, for such an author, there will be faster career growth in science, and they will be more invited to participate in scientific projects, etc. There is a "closed circle" effect here. If the author's publications are poorly cited, the career growth of such an author will be slower.

Since two performance assessment methods were used, the correlation coefficient between all assessments was calculated for their comparison. The correlation coefficient calculated between the estimates by the PageRank method and the number of citations equals 0.754. The correlation coefficient was also calculated for non-zero scores, equal to 0.647. This makes it possible to argue that the methods provide related but not functionally dependent estimates. Since relative evaluations are used for comparison, the different number of scientific publications from different patterns affects the evaluation result.

In many studies, for example in work [41], it is indicated that the participation of females in science is complicated, mainly due to pregnancy, the need to devote more time to raising children, and the greater representativeness of males in the management of scientific projects. Even a short-term pause in scientific activity can affect the dynamics of career growth in this direction, publication of high-quality scientific papers, research in scientific projects, etc. It can become more acute in different cultures and according to the socioeconomic status of the countries. Accordingly, this direction depends on ensuring gender equality in the country.

Based on the results, it can be concluded that scientific publications with male authors are cited more. Accordingly, their scientific publication productivity will be higher. It is established that the citation impact of a scientific publication depends on the gender composition of its authors. This means that the gender composition of scientific teams working on joint research affects their scientific publication productivity. Considering the superiority of publications with a male composition over publications with a female composition, we can conclude gender inequality. That is, the scientific publication produc-

tivity of female authors in these conditions will be lower than male authors. The results of this study confirm the results published in [42]. In particular, using coarsened exact matching, we show that publications by women are cited less by Wikipedia than expected, and publications by women are less likely to be cited than those by men.

However, the dynamics of evaluations of the advantages of subsets according to the defined patterns of the top ten countries by publication representation in the Citation Database Network Data show an overall improvement in gender equality in science.

Citation scores impacted scientific publications by certain countries' authors' gender parity scores, according to the Global Gender Gap Report 2022 [39]. It was established that the correlation coefficient is $-0.168$, which indicates a weak anti-correlation. This can be explained by the fact that the gender parity score refers to all aspects that affect gender equality in a country. In this study, only the aspect of scientific activity is considered, particularly one of its components: publication activity and citation of scientific publications. In addition, many other socioeconomic and cultural factors influence the equal representation of females and males in science and their scientific results.

*4.2. Limitations and Future Research Lines*

A limitation of the study is that in the Citation database Network Dataset, most publications relate to the subject area of natural sciences. Accordingly, the presentation of scientific publications in the social sciences or humanities could be more extensive. It is possible that, for publications in a non-naturalist subject area, value estimations of the citation impact of scientific publications will differ from those calculated in this research. Also, note that the number of citations to scientific publications in some countries may influence the received results. The presence of a small number of outliers in the dataset was established. However, based on the results of the calculation of statistical characteristics, it can be concluded that these emissions do not affect the PageRank score for countries with a sufficiently large amount of data. However, it can affect the calculation of the PageRank score of those countries for which there is insufficient data in the database.

The most common gender for a name may differ across countries. For example, Andrea is typically used for women in the U.S. and men in Italy. Such authors, taking into account the threshold value of the accuracy of the identification of the gender of the name, could be defined as authors with an unknown gender. In a future study, combining affiliation data with their gender imputation to improve accuracy will be used.

Another limitation is the impossibility of setting authors from non-binary gender since identifying whether the author is male or female was made based on their first names.

The more citations a given article receives over time, the higher its influence and the higher the author's productivity. Accordingly, one of the directions of future research is the assessment of aspects of the organization of project teams with different gender compositions on the productivity of each team member and the team's results as a whole. Also, an essential aspect of future research is to show the dynamics of changes in the evaluations of the preferences of subsets according to the corresponding patterns. In addition, the specified patterns can be considered patterns of scientific collaborations. This can be singled out as a separate indicator for assessing gender equality in scientific activity in different countries, regions, universities, etc. The research aims to inform countries, universities, and scientific institutes of problems related to gender gaps in science and to find ways to overcome them.

## 5. Conclusions

This work analyzed the citation impact of scientific publications by authors with different gender compositions. The PageRank method was used for citation impact evaluation of scientific publications and calculating the number of citations of scientific publications. The estimated citation impact of publications is calculated for different countries by eight subsets of publications that correspond to the patterns of the gender composition of their authors. The citation score is also calculated in cases where the gender composition of the

authors of a scientific publication cannot be identified. The advantages of evaluations for subsets corresponding to different patterns are calculated.

Based on the Citation Network Dataset, results of the citation impact evaluation of scientific publications with mostly male authors indicate that the citation impact of publications with a mixed gender composition prevails over the citation impact of publications with a only female composition. It indicates that articles from mainly male authors are cited more than articles with a mixed or female composition of authors. Analysis advantages in dynamics indicate that in the latter decade, there was a reduced influence of the gender composition of the authors' publications on citation impact. This may be the result of gender equality policies in many countries. However, the obtained results still confirm the existence of gender inequality in science, which may result from cultural and socioeconomic factors or natural homophily.

The obtained results can be considered more broadly. Author groups are often established, and the same author groups publish different publications in their direction. This means that citation scores obtained from calculation of the impact of scientific publications with different gender compositions of authors correspond to the assessment of the productivity of different gender patterns of scientists in scientific collaborations in different countries. This is important for intensifying the debate in the direction of ensuring gender equality and overcoming gender gaps in science. An increase in the scientific publishing activity of the authors contributes to the growth of the scientific productivity of the institutions with which these authors are affiliated. The obtained results do not indicate the presence of discrimination based on gender, and the results indicate the peculiarities of citing scientific publications with different gender compositions. However, the intensity of citations of such publications can be influenced by various socioeconomic, cultural, and other factors.

Appendix A (Tables A1–A3) presents the power of subsets of publications that correspond to the patterns of their gender composition. The average normalized PageRank scores indicated the citation impact of scientific publications by several citations for countries with more than 100 authors affiliated.

**Author Contributions:** Conceptualization and methodology, O.K. and Y.A.; software, Y.A.; analysis, O.K., Y.A., S.B. and A.F.; coding, Y.A.; Writing—Original draft preparation, O.K. and Y.A.; Writing— Review and editing, A.B., S.O. and A.M.; visualization O.K. and. Y.A.; project administration, A.B. All authors have read and agreed to the published version of the manuscript.

**Funding:** This research has been funded by the Committee of Science of the Ministry of Science and Higher Education of the Republic of Kazakhstan (Grant No.BR18574103 with the topic: "To increase the competitiveness of universities in Kazakhstan through the reengineering of the national system of quality assurance of higher education").

**Data Availability Statement:** All data are available in this publication. The data used to generate the figures in this article are available in Appendix A. Publicly available datasets analyzed in this study can be found here: Citation Network Dataset: DBLP + Citation, ACM Citation network. (2021). Aminer. Retrieved from: https://www.aminer.org/citation (accessed on 1 April 2023).

**Acknowledgments:** The authors thank the reviewers and editors for their generous and constructive comments that have improved this paper.

**Conflicts of Interest:** The authors declare no conflict of interest.

## Appendix A

**Table A1.** Power of subsets of posts that match patterns of their gender composition (data for countries with more than 100 authors).

| Country | Count | Pattern | | | | | | | | |
|---|---|---|---|---|---|---|---|---|---|---|
| | | N/A | Fff | Mff | Fmm | Mmm | Ffm | Mfm | F | M |
| USA | 442,281 | 7430 | 17,259 | 33,253 | 156,798 | 19,740 | 54,625 | 9153 | 45,685 | 98,338 |
| China | 412,520 | 5542 | 13,062 | 32,203 | 80,288 | 30,370 | 75,899 | 3298 | 9127 | 162,731 |
| Germany | 162,127 | 1167 | 4019 | 10,598 | 72,292 | 5175 | 18,475 | 3467 | 27,713 | 19,221 |
| France | 123,725 | 1633 | 4106 | 9972 | 42,829 | 6126 | 17,662 | 4410 | 18,075 | 18,912 |
| Japan | 110,524 | 412 | 1940 | 7775 | 59,387 | 2189 | 11,719 | 792 | 10,749 | 15,561 |
| Great Britain | 103,727 | 1311 | 3413 | 7782 | 34,887 | 4104 | 11,192 | 3186 | 15,937 | 21,915 |
| Italy | 98,243 | 2473 | 4456 | 9336 | 33,108 | 8485 | 19,035 | 1740 | 5824 | 13,786 |
| India | 96,816 | 2394 | 3830 | 8103 | 27,083 | 3443 | 8251 | 1007 | 4024 | 38,681 |
| Canada | 94,056 | 1546 | 3982 | 8290 | 36,520 | 3670 | 10,620 | 1547 | 7974 | 19,907 |
| Spain | 81,132 | 1157 | 2553 | 6638 | 29,979 | 5373 | 15,076 | 567 | 2824 | 16,965 |
| Australia | 59,920 | 973 | 2193 | 4850 | 21,038 | 2968 | 7780 | 1227 | 5736 | 13,155 |
| Taiwan | 59,137 | 323 | 961 | 1373 | 3992 | 527 | 1449 | 694 | 2040 | 47,778 |
| Brazil | 44,463 | 772 | 1730 | 3127 | 17,394 | 2659 | 7897 | 1188 | 3381 | 6315 |
| The Netherlands | 43,988 | 558 | 1270 | 3558 | 16,374 | 2274 | 5370 | 686 | 4152 | 9746 |
| Republic of Korea | 42,562 | 328 | 950 | 2653 | 14,760 | 919 | 3897 | 288 | 1569 | 17,198 |
| Iran | 32,109 | 354 | 1052 | 4127 | 13,627 | 1079 | 2789 | 201 | 1427 | 7453 |
| Singapore | 30,578 | 255 | 927 | 2246 | 8086 | 1197 | 3720 | 232 | 1109 | 12,806 |
| Hong Kong | 29,945 | 257 | 880 | 2107 | 7366 | 1091 | 3344 | 301 | 1263 | 13,336 |
| Poland | 29,603 | 530 | 1297 | 2217 | 12,600 | 850 | 2701 | 1108 | 6072 | 2228 |
| Switzerland | 29,296 | 237 | 768 | 2466 | 13,575 | 1194 | 4160 | 383 | 2728 | 3785 |
| Israel | 27,091 | 598 | 1320 | 2514 | 11,522 | 1006 | 3066 | 677 | 3067 | 3321 |
| Greece | 26,867 | 227 | 703 | 2220 | 12,430 | 986 | 3392 | 205 | 1594 | 5110 |
| Sweden | 26,577 | 519 | 952 | 2171 | 11,204 | 1148 | 3073 | 664 | 3159 | 3687 |
| Turkey | 26,471 | 794 | 1686 | 2484 | 9904 | 997 | 2297 | 622 | 2818 | 4869 |
| Austria | 25,093 | 229 | 637 | 1740 | 12,206 | 933 | 3152 | 382 | 2782 | 3032 |
| Belgium | 24,671 | 271 | 693 | 1935 | 10,513 | 1264 | 3647 | 335 | 2079 | 3934 |
| Finland | 22,618 | 604 | 722 | 1890 | 8364 | 1449 | 3286 | 598 | 2462 | 3243 |
| Portugal | 22,132 | 455 | 794 | 1897 | 10,002 | 1441 | 3376 | 250 | 1024 | 2893 |
| Georgia | 20,110 | 368 | 747 | 1516 | 7160 | 912 | 2593 | 426 | 1954 | 4434 |
| Russia | 18,801 | 279 | 794 | 1226 | 5293 | 719 | 2190 | 451 | 2465 | 5384 |
| Denmark | 15,055 | 250 | 454 | 1222 | 6412 | 679 | 2031 | 347 | 1941 | 1719 |
| Mexico | 15,044 | 169 | 486 | 1150 | 5567 | 680 | 2415 | 148 | 971 | 3458 |
| Czech Republic | 13,746 | 110 | 479 | 775 | 7105 | 251 | 1396 | 289 | 1942 | 1399 |
| Ireland | 13,360 | 181 | 434 | 1317 | 5644 | 694 | 1871 | 212 | 1072 | 1935 |
| Malaysia | 13,353 | 405 | 602 | 918 | 2845 | 925 | 1945 | 90 | 267 | 5356 |
| Norway | 13,206 | 246 | 457 | 1163 | 5291 | 553 | 1580 | 334 | 1629 | 1953 |
| New Zealand | 9889 | 158 | 416 | 900 | 3444 | 489 | 1306 | 211 | 1091 | 1874 |
| Pakistan | 9777 | 63 | 214 | 1057 | 4248 | 562 | 1570 | 40 | 286 | 1737 |
| Saudi Arabia | 8998 | 262 | 242 | 517 | 3542 | 234 | 675 | 147 | 1113 | 2266 |
| Hungary | 8487 | 48 | 274 | 523 | 4098 | 157 | 667 | 169 | 1490 | 1061 |
| Tunisia | 8475 | 528 | 243 | 2048 | 2057 | 1536 | 782 | 115 | 228 | 938 |
| Romania | 8429 | 262 | 494 | 948 | 2392 | 664 | 1097 | 336 | 1012 | 1224 |
| Egypt | 8042 | 123 | 291 | 758 | 2604 | 567 | 805 | 128 | 699 | 2067 |
| South Africa | 6947 | 206 | 365 | 544 | 2184 | 214 | 518 | 180 | 712 | 2024 |
| Chile | 6314 | 44 | 226 | 323 | 3385 | 238 | 910 | 36 | 395 | 757 |
| Algeria | 5849 | 197 | 253 | 891 | 2031 | 417 | 745 | 73 | 252 | 990 |
| Thailand | 5807 | 176 | 287 | 521 | 1141 | 250 | 441 | 128 | 311 | 2552 |
| Slovenia | 5032 | 96 | 231 | 491 | 2002 | 293 | 749 | 107 | 465 | 598 |
| Argentina | 4859 | 197 | 227 | 483 | 1634 | 466 | 808 | 76 | 261 | 707 |
| Morocco | 4659 | 89 | 67 | 769 | 1617 | 372 | 504 | 27 | 123 | 1091 |
| Serbia | 4445 | 171 | 222 | 468 | 1381 | 417 | 820 | 103 | 463 | 400 |
| Colombia | 4180 | 38 | 133 | 345 | 1795 | 231 | 766 | 30 | 152 | 690 |
| Vietnam | 4104 | 15 | 94 | 181 | 1243 | 75 | 352 | 38 | 255 | 1851 |
| UAE | 3895 | 49 | 153 | 430 | 1388 | 135 | 445 | 66 | 501 | 728 |
| Jordan | 3524 | 37 | 141 | 245 | 1497 | 126 | 494 | 56 | 452 | 476 |
| Croatia | 3334 | 113 | 186 | 339 | 1224 | 206 | 479 | 80 | 233 | 474 |
| Slovakia | 3129 | 88 | 276 | 330 | 1047 | 119 | 445 | 116 | 352 | 356 |
| Luxembourg | 3028 | 19 | 57 | 281 | 1449 | 117 | 547 | 40 | 225 | 293 |
| Cyprus | 2949 | 53 | 100 | 300 | 1301 | 122 | 335 | 43 | 258 | 437 |
| Bulgaria | 2690 | 155 | 197 | 329 | 510 | 198 | 348 | 174 | 353 | 426 |
| Qatar | 2467 | 24 | 62 | 267 | 1067 | 118 | 366 | 12 | 92 | 459 |

**Table A1.** *Cont.*

| Country | Count | Pattern | | | | | | | | |
|---|---|---|---|---|---|---|---|---|---|---|
| | | N/A | Fff | Mff | Fmm | Mmm | Ffm | Mfm | F | M |
| Bangladesh | 2275 | 34 | 38 | 95 | 332 | 77 | 124 | 18 | 55 | 1502 |
| Indonesia | 2266 | 71 | 102 | 203 | 549 | 152 | 283 | 31 | 83 | 792 |
| Lebanon | 2099 | 36 | 83 | 259 | 824 | 135 | 300 | 24 | 205 | 233 |
| Macedonia | 2058 | 55 | 113 | 254 | 760 | 168 | 328 | 28 | 126 | 226 |
| Peru | 2049 | 47 | 98 | 170 | 786 | 133 | 387 | 39 | 106 | 283 |
| Ukraine | 1981 | 47 | 89 | 106 | 509 | 142 | 367 | 29 | 245 | 447 |
| Estonia | 1822 | 35 | 67 | 179 | 727 | 107 | 237 | 59 | 210 | 201 |
| Lithuania | 1768 | 44 | 106 | 135 | 639 | 79 | 244 | 40 | 246 | 235 |
| Kuwait | 1405 | 32 | 55 | 84 | 521 | 22 | 124 | 38 | 260 | 269 |
| Latvia | 1251 | 102 | 116 | 92 | 281 | 86 | 123 | 80 | 179 | 192 |
| Ecuador | 1190 | 18 | 35 | 106 | 412 | 116 | 309 | 6 | 36 | 152 |
| Philippines | 1046 | 31 | 58 | 109 | 266 | 99 | 189 | 37 | 65 | 192 |
| Niger | 1041 | 13 | 39 | 63 | 325 | 35 | 111 | 17 | 140 | 298 |
| Nigeria | 1032 | 13 | 38 | 63 | 319 | 35 | 111 | 17 | 138 | 298 |
| Mongolia | 968 | 19 | 25 | 86 | 176 | 72 | 164 | 19 | 31 | 376 |
| Iraq | 958 | 22 | 38 | 56 | 411 | 37 | 130 | 12 | 86 | 166 |
| Cuba | 943 | 22 | 30 | 85 | 306 | 88 | 174 | 6 | 20 | 212 |
| Venezuela | 936 | 28 | 51 | 85 | 298 | 49 | 103 | 12 | 71 | 239 |
| Uruguay | 887 | 18 | 36 | 68 | 405 | 56 | 122 | 14 | 72 | 96 |
| Iceland | 808 | 17 | 35 | 44 | 340 | 44 | 142 | 17 | 75 | 94 |
| Montenegro | 718 | 27 | 36 | 69 | 196 | 36 | 101 | 23 | 132 | 98 |
| Oman | 704 | 2 | 25 | 46 | 269 | 11 | 40 | 16 | 116 | 179 |
| Malta | 687 | 2 | 30 | 70 | 324 | 21 | 84 | 7 | 73 | 76 |
| Sri Lanka | 620 | 17 | 15 | 59 | 84 | 38 | 48 | 4 | 15 | 340 |
| Kazakhstan | 607 | 22 | 21 | 52 | 129 | 57 | 74 | 3 | 72 | 177 |
| Macau | 582 | 8 | 19 | 41 | 148 | 23 | 69 | 18 | 17 | 239 |
| Belarus | 572 | 7 | 26 | 43 | 153 | 5 | 69 | 11 | 53 | 205 |
| Puerto Rico | 483 | 5 | 14 | 27 | 178 | 21 | 63 | 5 | 52 | 118 |
| Saint Martin | 445 | 14 | 11 | 25 | 172 | 32 | 52 | 12 | 42 | 85 |
| Ethiopia | 380 | 1 | 12 | 25 | 168 | 3 | 43 | 3 | 35 | 90 |
| Small | 364 | 1 | 11 | 17 | 122 | 10 | 37 | 4 | 42 | 120 |
| Kenya | 324 | 3 | 19 | 22 | 100 | 24 | 62 | 7 | 23 | 64 |
| Armenia | 318 | 5 | 15 | 16 | 99 | 6 | 34 | 9 | 40 | 94 |
| Cameroon | 315 | 2 | 11 | 13 | 94 | 14 | 36 | 5 | 18 | 122 |
| Azerbaijan | 310 | 7 | 13 | 7 | 81 | 7 | 23 | 3 | 37 | 132 |
| Bosnia and Herzegovina | 302 | 13 | 17 | 35 | 89 | 29 | 51 | 6 | 24 | 38 |
| Palestine | 301 | 0 | 12 | 17 | 138 | 3 | 40 | 2 | 57 | 32 |
| Ghana | 299 | 1 | 8 | 20 | 146 | 1 | 36 | 3 | 37 | 47 |
| Costa Rica | 265 | 10 | 14 | 19 | 99 | 14 | 45 | 5 | 31 | 28 |
| Bahrain | 247 | 6 | 5 | 17 | 62 | 8 | 17 | 12 | 55 | 65 |
| Senegal | 194 | 0 | 6 | 20 | 72 | 7 | 28 | 0 | 12 | 49 |
| Brunei | 193 | 1 | 8 | 6 | 33 | 10 | 18 | 3 | 13 | 101 |
| Uganda | 187 | 2 | 6 | 25 | 59 | 13 | 32 | 8 | 14 | 28 |
| Myanmar | 187 | 29 | 26 | 30 | 17 | 8 | 7 | 2 | 8 | 60 |
| Mauritius | 184 | 7 | 6 | 18 | 23 | 6 | 6 | 1 | 8 | 109 |
| Libya | 171 | 1 | 7 | 10 | 65 | 4 | 11 | 2 | 18 | 53 |
| Fiji | 168 | 0 | 11 | 12 | 60 | 9 | 23 | 4 | 17 | 32 |
| Panama | 167 | 4 | 9 | 13 | 67 | 8 | 29 | 4 | 11 | 22 |
| Paraguay | 161 | 0 | 2 | 16 | 91 | 9 | 28 | 0 | 4 | 11 |
| Jamaica | 157 | 1 | 13 | 16 | 44 | 3 | 20 | 8 | 16 | 36 |
| Albania | 150 | 2 | 5 | 22 | 39 | 17 | 37 | 3 | 4 | 21 |
| Tanzania | 144 | 1 | 8 | 12 | 43 | 7 | 13 | 3 | 23 | 34 |
| Benin | 138 | 3 | 6 | 14 | 38 | 2 | 20 | 3 | 14 | 38 |
| Moldova | 134 | 7 | 6 | 2 | 64 | 6 | 16 | 0 | 18 | 15 |
| Liechtenstein | 125 | 0 | 3 | 10 | 64 | 3 | 22 | 1 | 11 | 11 |
| Yemale | 118 | 0 | 6 | 10 | 42 | 0 | 13 | 1 | 16 | 30 |
| Botswana | 117 | 0 | 6 | 5 | 35 | 2 | 6 | 1 | 7 | 55 |
| Sudan | 112 | 4 | 2 | 24 | 33 | 4 | 14 | 4 | 7 | 20 |
| Namibia | 111 | 12 | 7 | 16 | 15 | 9 | 18 | 9 | 10 | 15 |
| Syria | 105 | 2 | 2 | 18 | 39 | 3 | 10 | 0 | 11 | 20 |
| Trinidad and Tobago | 102 | 1 | 4 | 15 | 30 | 1 | 2 | 11 | 21 | 17 |

**Table A2.** Average normalized PageRank scores citation impact of scientific publications for countries with which more than 100 authors are affiliated.

| Country | Count | Pattern | | | | | | | | |
|---|---|---|---|---|---|---|---|---|---|---|
| | | N/A | Fff | Mff | Fmm | Mmm | Ffm | Mfm | F | M |
| USA | 442,281 | 1.000 | 0.515 | 0.714 | 0.732 | 0.881 | 0.562 | 0.727 | 0.460 | 0.775 |
| China | 412,520 | 0.989 | 0.676 | 0.786 | 0.809 | 1.000 | 0.669 | 0.829 | 0.758 | 0.970 |
| Germany | 162,127 | 1.000 | 0.462 | 0.528 | 0.570 | 0.722 | 0.510 | 0.649 | 0.301 | 0.377 |
| France | 123,725 | 1.000 | 0.876 | 0.892 | 0.694 | 0.923 | 0.600 | 0.737 | 0.299 | 0.497 |
| Japan | 110,524 | 1.000 | 0.602 | 0.757 | 0.629 | 0.745 | 0.482 | 0.702 | 0.709 | 0.764 |
| Great Britain | 103,727 | 1.000 | 0.957 | 0.836 | 0.861 | 0.997 | 0.818 | 0.878 | 0.476 | 0.582 |
| Italy | 98,243 | 1.000 | 0.531 | 0.646 | 0.638 | 0.821 | 0.546 | 0.642 | 0.513 | 0.775 |
| India | 96,816 | 0.882 | 0.568 | 0.739 | 0.595 | 0.835 | 0.639 | 0.778 | 0.736 | 1.000 |
| Canada | 94,056 | 1.000 | 0.711 | 0.668 | 0.642 | 0.836 | 0.605 | 0.719 | 0.578 | 0.939 |
| Spain | 81,132 | 1.000 | 0.672 | 0.874 | 0.784 | 0.931 | 0.675 | 0.832 | 0.603 | 0.771 |
| Australia | 59,920 | 1.000 | 0.812 | 0.687 | 0.708 | 0.946 | 0.658 | 0.768 | 0.649 | 0.797 |
| Taiwan | 59,137 | 0.581 | 0.499 | 0.564 | 0.491 | 1.000 | 0.374 | 0.439 | 0.560 | 0.609 |
| Brazil | 44,463 | 1.000 | 0.662 | 0.655 | 0.763 | 0.838 | 0.655 | 0.741 | 0.196 | 0.296 |
| The Netherlands | 43,988 | 1.000 | 0.453 | 0.675 | 0.616 | 0.761 | 0.502 | 0.648 | 0.453 | 0.728 |
| Republic of Korea | 42,562 | 1.000 | 0.445 | 0.602 | 0.479 | 0.780 | 0.502 | 0.636 | 0.334 | 0.737 |
| Iran | 32,109 | 1.000 | 0.669 | 0.833 | 0.716 | 0.784 | 0.722 | 0.805 | 0.603 | 0.575 |
| Singapore | 30,578 | 1.000 | 0.461 | 0.508 | 0.521 | 0.633 | 0.415 | 0.550 | 0.453 | 0.629 |
| Hong Kong | 29,945 | 1.000 | 0.543 | 0.732 | 0.661 | 0.884 | 0.462 | 0.802 | 0.436 | 0.700 |
| Poland | 29,603 | 1.000 | 0.426 | 0.595 | 0.750 | 0.802 | 0.468 | 0.626 | 0.524 | 0.731 |
| Switzerland | 29,296 | 1.000 | 0.653 | 0.591 | 0.710 | 0.765 | 0.500 | 0.580 | 0.904 | 0.692 |
| Israel | 27,091 | 1.000 | 0.435 | 0.540 | 0.533 | 0.721 | 0.442 | 0.600 | 0.620 | 0.649 |
| Greece | 26,867 | 1.000 | 0.716 | 0.937 | 0.688 | 0.767 | 0.656 | 0.784 | 0.874 | 0.886 |
| Sweden | 26,577 | 1.000 | 0.512 | 0.589 | 0.557 | 0.713 | 0.480 | 0.585 | 0.456 | 0.693 |
| Turkey | 26,471 | 1.000 | 0.517 | 0.701 | 0.618 | 0.781 | 0.605 | 0.609 | 0.456 | 0.673 |
| Austria | 25,093 | 1.000 | 0.495 | 0.772 | 0.619 | 0.783 | 0.599 | 0.710 | 0.525 | 0.752 |
| Belgium | 24,671 | 1.000 | 0.486 | 0.866 | 0.854 | 0.918 | 0.568 | 0.745 | 0.555 | 0.767 |
| Finland | 22,618 | 0.852 | 0.416 | 0.621 | 0.537 | 0.867 | 0.542 | 0.720 | 0.494 | 1.000 |
| Portugal | 22,132 | 0.916 | 0.639 | 1.000 | 0.727 | 0.732 | 0.766 | 0.677 | 0.732 | 0.982 |
| Georgia | 20,110 | 0.949 | 0.653 | 0.990 | 0.802 | 0.937 | 0.745 | 0.783 | 0.781 | 1.000 |
| Russia | 18,801 | 0.798 | 0.544 | 0.688 | 1.000 | 0.742 | 0.537 | 0.631 | 0.636 | 0.784 |
| Denmark | 15,055 | 0.767 | 0.490 | 0.754 | 0.621 | 0.916 | 0.565 | 0.961 | 0.382 | 1.000 |
| Mexico | 15,044 | 1.000 | 0.422 | 0.444 | 0.452 | 0.507 | 0.401 | 0.492 | 0.260 | 0.457 |
| Czech Republic | 13,746 | 1.000 | 0.403 | 0.812 | 0.689 | 0.830 | 0.497 | 0.720 | 0.412 | 0.695 |
| Ireland | 13,360 | 0.565 | 0.349 | 1.000 | 0.498 | 0.528 | 0.355 | 0.996 | 0.282 | 0.468 |
| Malaysia | 13,353 | 0.718 | 0.168 | 0.709 | 0.436 | 0.588 | 0.348 | 0.534 | 0.427 | 1.000 |
| Norway | 13,206 | 1.000 | 0.343 | 0.633 | 0.499 | 0.738 | 0.553 | 0.682 | 0.429 | 0.626 |
| New Zealand | 9889 | 1.000 | 0.450 | 0.638 | 0.689 | 0.878 | 0.542 | 0.851 | 0.463 | 0.923 |
| Pakistan | 9777 | 1.000 | 0.450 | 0.449 | 0.302 | 0.482 | 0.367 | 0.376 | 0.230 | 0.513 |
| Saudi Arabia | 8998 | 0.766 | 0.578 | 0.621 | 0.663 | 1.000 | 0.529 | 0.550 | 0.509 | 0.835 |
| Hungary | 8487 | 1.000 | 0.368 | 0.578 | 0.408 | 0.795 | 0.375 | 0.559 | 0.360 | 0.759 |
| Tunisia | 8475 | 1.000 | 0.511 | 0.593 | 0.634 | 0.839 | 0.631 | 0.744 | 0.528 | 0.840 |
| Romania | 8429 | 0.657 | 0.226 | 1.000 | 0.364 | 0.649 | 0.408 | 0.486 | 0.226 | 0.663 |
| Egypt | 8042 | 1.000 | 0.342 | 0.439 | 0.592 | 0.732 | 0.447 | 0.556 | 0.405 | 0.662 |
| South Africa | 6947 | 0.899 | 0.643 | 0.619 | 0.824 | 0.877 | 0.674 | 1.000 | 0.313 | 0.605 |
| Chile | 6314 | 0.772 | 0.655 | 0.582 | 0.587 | 0.814 | 0.503 | 0.622 | 0.479 | 1.000 |
| Algeria | 5849 | 0.981 | 0.341 | 0.892 | 0.674 | 0.817 | 0.453 | 0.843 | 0.100 | 1.000 |
| Thailand | 5807 | 1.000 | 0.334 | 0.370 | 0.900 | 0.975 | 0.449 | 0.766 | 0.364 | 0.755 |
| Slovenia | 5032 | 1.000 | 0.575 | 0.758 | 0.769 | 0.972 | 0.699 | 0.795 | 0.563 | 0.832 |
| Argentina | 4859 | 1.000 | 0.520 | 0.939 | 0.609 | 0.905 | 0.405 | 0.820 | 0.678 | 0.905 |
| Morocco | 4659 | 0.297 | 1.000 | 0.208 | 0.157 | 0.311 | 0.121 | 0.199 | 0.203 | 0.627 |
| Serbia | 4445 | 0.573 | 0.725 | 0.754 | 0.677 | 0.792 | 0.636 | 0.795 | 1.000 | 0.596 |
| Colombia | 4180 | 1.000 | 0.269 | 0.483 | 0.549 | 0.478 | 0.454 | 0.438 | 0.811 | 0.760 |
| Vietnam | 4104 | 1.000 | 0.471 | 0.409 | 0.734 | 0.925 | 0.578 | 0.563 | 0.500 | 0.834 |
| UAE | 3895 | 0.747 | 0.371 | 0.661 | 0.684 | 1.000 | 0.690 | 0.952 | 0.422 | 0.735 |
| Jordan | 3524 | 0.769 | 0.325 | 0.529 | 0.705 | 1.000 | 0.413 | 0.745 | 0.586 | 0.385 |
| Croatia | 3334 | 1.000 | 0.236 | 0.342 | 0.507 | 0.708 | 0.333 | 0.518 | 0.137 | 0.696 |
| Slovakia | 3129 | 0.794 | 0.406 | 0.507 | 0.595 | 0.481 | 0.419 | 0.495 | 1.000 | 0.505 |
| Luxembourg | 3028 | 1.000 | 0.566 | 0.706 | 0.969 | 0.639 | 0.301 | 0.499 | 0.694 | 0.748 |
| Cyprus | 2949 | 0.987 | 0.543 | 0.518 | 0.729 | 0.881 | 0.947 | 0.775 | 0.564 | 1.000 |
| Bulgaria | 2690 | 1.000 | 0.483 | 0.422 | 0.378 | 0.693 | 0.303 | 0.670 | 0.538 | 0.629 |
| Qatar | 2467 | 1.000 | 0.370 | 0.391 | 0.906 | 0.621 | 0.679 | 0.635 | 0.372 | 0.743 |
| Bangladesh | 2275 | 0.870 | 0.175 | 0.650 | 1.000 | 0.723 | 0.438 | 0.672 | 0.428 | 0.524 |
| Indonesia | 2266 | 0.590 | 0.108 | 0.214 | 0.299 | 0.540 | 0.263 | 0.288 | 0.597 | 1.000 |
| Lebanon | 2099 | 0.868 | 0.140 | 0.393 | 0.298 | 1.000 | 0.464 | 0.562 | 0.846 | 0.672 |
| Macedonia | 2058 | 1.000 | 0.234 | 0.522 | 0.657 | 0.792 | 0.761 | 0.863 | 0.118 | 0.926 |

**Table A2.** *Cont.*

| Country | Count | Pattern | | | | | | | | |
|---|---|---|---|---|---|---|---|---|---|---|
| | | N/A | Fff | Mff | Fmm | Mmm | Ffm | Mfm | F | M |
| Peru | 2049 | 0.724 | 0.575 | 0.653 | 0.450 | 0.803 | 0.431 | 0.620 | 0.042 | 1.000 |
| Ukraine | 1981 | 0.996 | 0.621 | 0.324 | 0.468 | 1.000 | 0.818 | 0.654 | 0.640 | 0.533 |
| Estonia | 1822 | 0.900 | 0.050 | 0.198 | 1.000 | 0.604 | 0.839 | 0.850 | 0.316 | 0.715 |
| Lithuania | 1768 | 0.789 | 0.240 | 0.777 | 0.448 | 1.000 | 0.421 | 0.534 | 0.327 | 0.949 |
| Kuwait | 1405 | 0.936 | 0.548 | 0.439 | 0.174 | 1.000 | 0.419 | 0.442 | 0.182 | 0.545 |
| Latvia | 1251 | 0.255 | 1.000 | 0.297 | 0.167 | 0.164 | 0.131 | 0.134 | 0.029 | 0.146 |
| Ecuador | 1190 | 1.000 | 0.458 | 0.197 | 0.571 | 0.551 | 0.602 | 0.612 | 0.000 | 0.084 |
| Philippines | 1046 | 0.720 | 0.000 | 0.380 | 0.525 | 0.516 | 0.000 | 0.827 | 1.000 | 0.466 |
| Niger | 1041 | 1.000 | 0.135 | 0.758 | 0.262 | 0.885 | 0.156 | 0.368 | 0.138 | 0.920 |
| Nigeria | 1032 | 1.000 | 0.039 | 0.457 | 0.073 | 0.453 | 0.341 | 0.588 | 0.073 | 0.467 |
| Mongolia | 968 | 0.694 | 0.354 | 0.286 | 0.310 | 0.622 | 0.155 | 0.521 | 1.000 | 0.153 |
| Iraq | 958 | 0.649 | 0.000 | 1.000 | 0.236 | 0.185 | 0.144 | 0.224 | 0.623 | 0.078 |
| Cuba | 943 | 0.268 | 1.000 | 0.279 | 0.151 | 0.499 | 0.083 | 0.355 | 0.536 | 0.258 |
| Venezuela | 936 | 0.366 | 0.000 | 1.000 | 0.319 | 0.533 | 0.358 | 0.603 | 0.000 | 0.010 |
| Uruguay | 887 | 0.648 | 0.129 | 0.606 | 1.000 | 0.439 | 0.828 | 0.584 | 0.670 | 0.454 |
| Iceland | 808 | 0.159 | 0.000 | 1.000 | 0.094 | 0.187 | 0.128 | 0.240 | 0.009 | 0.110 |
| Montenegro | 718 | 0.655 | 0.000 | 0.960 | 0.484 | 1.000 | 0.551 | 0.602 | 0.000 | 0.202 |
| Oman | 704 | 0.801 | 0.000 | 0.076 | 1.000 | 0.263 | 0.012 | 0.622 | 0.140 | 0.332 |
| Malta | 687 | 0.191 | 0.000 | 0.000 | 0.572 | 0.966 | 0.448 | 1.000 | 0.031 | 0.062 |
| Sri Lanka | 620 | 0.857 | 0.000 | 0.618 | 0.157 | 0.681 | 0.023 | 0.159 | 0.000 | 1.000 |
| Kazakhstan | 607 | 0.297 | 0.589 | 0.000 | 0.722 | 1.000 | 0.231 | 0.256 | 0.249 | 0.497 |
| Macau | 582 | 0.280 | 0.165 | 0.418 | 0.932 | 0.128 | 0.322 | 0.665 | 0.177 | 1.000 |
| Belarus | 572 | 0.427 | 0.051 | 0.000 | 0.260 | 1.000 | 0.048 | 0.117 | 0.000 | 0.694 |
| Puerto Rico | 483 | 0.662 | 1.000 | 0.452 | 0.000 | 0.480 | 0.347 | 0.351 | 0.011 | 0.582 |
| Saint Martin | 445 | 1.000 | 0.132 | 0.000 | 0.304 | 0.297 | 0.101 | 0.489 | 0.000 | 0.244 |
| Ethiopia | 380 | 0.524 | 0.000 | 0.000 | 0.080 | 0.168 | 1.000 | 0.073 | 0.000 | 0.075 |
| Small | 364 | 0.906 | 0.000 | 0.000 | 1.000 | 0.693 | 0.000 | 0.855 | 0.000 | 0.090 |
| Kenya | 324 | 0.179 | 0.000 | 0.000 | 0.000 | 0.332 | 0.852 | 1.000 | 0.000 | 0.000 |
| Armenia | 318 | 0.651 | 0.000 | 0.000 | 0.250 | 0.216 | 0.000 | 1.000 | 0.000 | 0.608 |
| Cameroon | 315 | 0.219 | 0.000 | 0.000 | 0.000 | 0.241 | 1.000 | 0.000 | 0.000 | 0.142 |
| Azerbaijan | 310 | 0.000 | 0.000 | 0.017 | 0.030 | 0.134 | 0.000 | 1.000 | 0.000 | 0.000 |
| Bosnia and Herzegovina | 302 | 0.000 | 0.000 | 0.000 | 0.000 | 1.000 | 0.000 | 0.000 | 0.000 | 0.000 |
| Palestine | 301 | 0.292 | 0.000 | 1.000 | 0.000 | 0.005 | 0.211 | 0.309 | 0.000 | 0.000 |
| Ghana | 299 | 0.361 | 0.000 | 0.000 | 0.283 | 1.000 | 0.000 | 0.827 | 0.000 | 0.000 |
| Costa Rica | 265 | 0.092 | 0.000 | 1.000 | 0.000 | 0.026 | 0.000 | 0.000 | 0.000 | 0.138 |
| Bahrain | 247 | 0.567 | 0.000 | 0.000 | 0.000 | 1.000 | 0.000 | 0.000 | 0.000 | 0.000 |
| Senegal | 194 | 1.000 | 0.000 | 0.000 | 0.000 | 0.000 | 0.000 | 0.000 | 0.000 | 0.000 |
| Brunei | 193 | 1.000 | 0.000 | 0.000 | 0.000 | 0.000 | 0.000 | 0.000 | 0.000 | 0.000 |
| Uganda | 187 | 1.000 | 0.515 | 0.714 | 0.732 | 0.881 | 0.562 | 0.727 | 0.460 | 0.775 |
| Myanmar | 187 | 0.989 | 0.676 | 0.786 | 0.809 | 1.000 | 0.669 | 0.829 | 0.758 | 0.970 |
| Mauritius | 184 | 1.000 | 0.462 | 0.528 | 0.570 | 0.722 | 0.510 | 0.649 | 0.301 | 0.377 |
| Libya | 171 | 1.000 | 0.876 | 0.892 | 0.694 | 0.923 | 0.600 | 0.737 | 0.299 | 0.497 |
| Fiji | 168 | 1.000 | 0.602 | 0.757 | 0.629 | 0.745 | 0.482 | 0.702 | 0.709 | 0.764 |
| Panama | 167 | 1.000 | 0.957 | 0.836 | 0.861 | 0.997 | 0.818 | 0.878 | 0.476 | 0.582 |
| Paraguay | 161 | 1.000 | 0.531 | 0.646 | 0.638 | 0.821 | 0.546 | 0.642 | 0.513 | 0.775 |
| Jamaica | 157 | 0.882 | 0.568 | 0.739 | 0.595 | 0.835 | 0.639 | 0.778 | 0.736 | 1.000 |
| Albania | 150 | 1.000 | 0.711 | 0.668 | 0.642 | 0.836 | 0.605 | 0.719 | 0.578 | 0.939 |
| Tanzania | 144 | 1.000 | 0.672 | 0.874 | 0.784 | 0.931 | 0.675 | 0.832 | 0.603 | 0.771 |
| Benin | 138 | 1.000 | 0.812 | 0.687 | 0.708 | 0.946 | 0.658 | 0.768 | 0.649 | 0.797 |
| Moldova | 134 | 0.581 | 0.499 | 0.564 | 0.491 | 1.000 | 0.374 | 0.439 | 0.560 | 0.609 |
| Liechtenstein | 125 | 1.000 | 0.662 | 0.655 | 0.763 | 0.838 | 0.655 | 0.741 | 0.196 | 0.296 |
| Yemale | 118 | 1.000 | 0.453 | 0.675 | 0.616 | 0.761 | 0.502 | 0.648 | 0.453 | 0.728 |
| Botswana | 117 | 1.000 | 0.445 | 0.602 | 0.479 | 0.780 | 0.502 | 0.636 | 0.334 | 0.737 |
| Sudan | 112 | 1.000 | 0.669 | 0.833 | 0.716 | 0.784 | 0.722 | 0.805 | 0.603 | 0.575 |
| Namibia | 111 | 1.000 | 0.461 | 0.508 | 0.521 | 0.633 | 0.415 | 0.550 | 0.453 | 0.629 |
| Syria | 105 | 1.000 | 0.543 | 0.732 | 0.661 | 0.884 | 0.462 | 0.802 | 0.436 | 0.700 |
| Trinidad and Tobago | 102 | 1.000 | 0.426 | 0.595 | 0.750 | 0.802 | 0.468 | 0.626 | 0.524 | 0.731 |

**Table A3.** Average normalized estimates of citation impact by the number of citations of scientific publications for countries with which more than 100 authors are affiliated.

| Country | Count | Pattern | | | | | | | | |
|---|---|---|---|---|---|---|---|---|---|---|
| | | N/A | Fff | Mff | Fmm | Mmm | Ffm | Mfm | F | M |
| USA | 442,281 | 1.000 | 0.542 | 0.729 | 0.778 | 0.946 | 0.662 | 0.871 | 0.386 | 0.647 |
| China | 412,520 | 1.000 | 0.620 | 0.710 | 0.759 | 0.993 | 0.641 | 0.832 | 0.613 | 0.709 |
| Germany | 162,127 | 1.000 | 0.453 | 0.536 | 0.653 | 0.842 | 0.653 | 0.846 | 0.267 | 0.352 |
| France | 123,725 | 0.892 | 0.428 | 0.887 | 0.676 | 0.902 | 0.637 | 0.741 | 0.252 | 1.000 |
| Japan | 110,524 | 1.000 | 0.550 | 0.652 | 0.617 | 0.766 | 0.507 | 0.772 | 0.618 | 0.642 |
| Great Britain | 103,727 | 0.911 | 0.878 | 0.801 | 0.808 | 1.000 | 0.828 | 0.933 | 0.378 | 0.525 |
| Italy | 98,243 | 1.000 | 0.565 | 0.700 | 0.672 | 0.904 | 0.596 | 0.698 | 0.511 | 0.785 |
| India | 96,816 | 0.875 | 0.514 | 0.756 | 0.596 | 0.897 | 0.712 | 0.861 | 0.668 | 1.000 |
| Canada | 94,056 | 1.000 | 0.757 | 0.677 | 0.671 | 0.901 | 0.709 | 0.863 | 0.577 | 0.874 |
| Spain | 81,132 | 1.000 | 0.643 | 0.909 | 0.817 | 0.934 | 0.757 | 0.841 | 0.436 | 0.713 |
| Australia | 59,920 | 1.000 | 0.801 | 0.656 | 0.681 | 1.000 | 0.707 | 0.811 | 0.470 | 0.757 |
| Taiwan | 59,137 | 0.581 | 0.440 | 0.570 | 0.477 | 1.000 | 0.417 | 0.464 | 0.457 | 0.544 |
| Brazil | 44,463 | 1.000 | 0.563 | 0.596 | 0.733 | 0.836 | 0.640 | 0.735 | 0.132 | 0.259 |
| The Netherlands | 43,988 | 1.000 | 0.493 | 0.741 | 0.678 | 0.803 | 0.557 | 0.706 | 0.433 | 0.707 |
| Republic of Korea | 42,562 | 1.000 | 0.623 | 0.723 | 0.700 | 0.904 | 0.877 | 1.000 | 0.343 | 0.657 |
| Iran | 32,109 | 1.000 | 0.315 | 0.526 | 0.431 | 0.718 | 0.551 | 0.609 | 0.253 | 0.544 |
| Singapore | 30,578 | 1.000 | 0.694 | 0.840 | 0.716 | 0.741 | 0.744 | 0.837 | 0.566 | 0.447 |
| Hong Kong | 29,945 | 1.000 | 0.597 | 0.867 | 0.876 | 1.000 | 0.704 | 0.955 | 0.364 | 0.547 |
| Poland | 29,603 | 1.000 | 0.340 | 0.408 | 0.485 | 0.643 | 0.375 | 0.575 | 0.315 | 0.497 |
| Switzerland | 29,296 | 1.000 | 0.520 | 0.753 | 0.700 | 0.954 | 0.532 | 1.000 | 0.380 | 0.730 |
| Israel | 27,091 | 1.000 | 0.449 | 0.623 | 0.913 | 0.952 | 0.615 | 0.815 | 0.481 | 0.685 |
| Greece | 26,867 | 1.000 | 0.670 | 0.603 | 0.746 | 0.853 | 0.523 | 0.583 | 0.472 | 0.635 |
| Sweden | 26,577 | 1.000 | 0.421 | 0.606 | 0.590 | 0.777 | 0.446 | 0.787 | 0.558 | 0.599 |
| Turkey | 26,471 | 1.000 | 0.816 | 0.853 | 0.674 | 0.787 | 0.752 | 0.805 | 0.679 | 0.722 |
| Austria | 25,093 | 1.000 | 0.501 | 0.714 | 0.685 | 0.867 | 0.649 | 0.762 | 0.428 | 0.672 |
| Belgium | 24,671 | 1.000 | 0.517 | 0.763 | 0.661 | 0.849 | 0.616 | 0.642 | 0.336 | 0.564 |
| Finland | 22,618 | 0.852 | 0.552 | 0.759 | 0.638 | 0.784 | 0.647 | 0.819 | 0.913 | 0.740 |
| Portugal | 22,132 | 0.916 | 0.487 | 0.810 | 0.901 | 0.973 | 0.546 | 0.760 | 0.496 | 0.794 |
| Georgia | 20,110 | 0.949 | 0.671 | 0.830 | 0.785 | 0.818 | 0.441 | 0.791 | 0.200 | 0.467 |
| Russia | 18,801 | 0.798 | 0.401 | 0.414 | 0.535 | 0.817 | 0.480 | 1.000 | 0.266 | 0.599 |
| Denmark | 15,055 | 0.767 | 0.556 | 1.000 | 0.736 | 0.964 | 0.675 | 0.805 | 0.650 | 0.890 |
| Mexico | 15,044 | 1.000 | 1.000 | 0.925 | 0.724 | 0.665 | 0.759 | 0.622 | 0.533 | 0.688 |
| Czech Republic | 13,746 | 1.000 | 0.240 | 0.700 | 1.000 | 0.752 | 0.483 | 0.657 | 0.393 | 0.620 |
| Ireland | 13,360 | 0.565 | 0.592 | 0.714 | 0.642 | 0.774 | 0.636 | 0.738 | 0.305 | 0.588 |
| Malaysia | 13,353 | 0.718 | 0.557 | 0.871 | 0.656 | 0.916 | 0.603 | 0.968 | 0.286 | 1.000 |
| Norway | 13,206 | 1.000 | 0.301 | 0.932 | 0.729 | 0.961 | 0.532 | 0.838 | 0.276 | 0.595 |
| New Zealand | 9889 | 1.000 | 0.120 | 1.000 | 0.211 | 0.201 | 0.135 | 0.395 | 0.095 | 0.140 |
| Pakistan | 9777 | 1.000 | 0.142 | 0.807 | 0.501 | 0.718 | 0.373 | 0.629 | 0.254 | 1.000 |
| Saudi Arabia | 8998 | 0.766 | 0.222 | 0.547 | 0.457 | 0.694 | 0.505 | 0.655 | 0.275 | 0.440 |
| Hungary | 8487 | 1.000 | 0.554 | 0.502 | 0.404 | 0.584 | 0.770 | 0.566 | 0.173 | 0.442 |
| Tunisia | 8475 | 1.000 | 0.287 | 0.615 | 0.677 | 0.831 | 0.542 | 0.909 | 0.323 | 0.766 |
| Romania | 8429 | 0.657 | 0.420 | 0.493 | 0.585 | 1.000 | 0.478 | 0.524 | 0.389 | 0.657 |
| Egypt | 8042 | 1.000 | 0.444 | 0.644 | 0.435 | 0.919 | 0.424 | 0.653 | 0.317 | 0.718 |
| South Africa | 6947 | 0.899 | 0.767 | 0.510 | 0.589 | 0.852 | 0.602 | 0.764 | 0.446 | 0.646 |
| Chile | 6314 | 0.772 | 0.055 | 1.000 | 0.208 | 0.454 | 0.183 | 0.346 | 0.104 | 0.484 |
| Algeria | 5849 | 0.981 | 0.270 | 0.360 | 0.525 | 0.685 | 0.389 | 0.469 | 0.129 | 0.453 |
| Thailand | 5807 | 1.000 | 0.555 | 0.568 | 0.809 | 0.920 | 0.891 | 0.954 | 0.261 | 0.622 |
| Slovenia | 5032 | 1.000 | 0.376 | 0.529 | 0.825 | 0.762 | 0.640 | 0.633 | 0.462 | 0.667 |
| Argentina | 4859 | 1.000 | 0.678 | 0.699 | 0.719 | 1.000 | 0.607 | 0.774 | 0.413 | 0.798 |
| Morocco | 4659 | 0.297 | 0.194 | 0.463 | 0.488 | 0.736 | 0.399 | 1.000 | 0.008 | 0.772 |
| Serbia | 4445 | 0.573 | 0.367 | 0.485 | 0.683 | 0.811 | 0.681 | 0.708 | 0.312 | 0.822 |
| Colombia | 4180 | 1.000 | 0.228 | 0.265 | 0.641 | 0.805 | 0.390 | 0.680 | 0.212 | 0.338 |
| Vietnam | 4104 | 1.000 | 0.288 | 0.415 | 0.699 | 0.966 | 0.596 | 0.812 | 0.267 | 0.535 |
| UAE | 3895 | 0.747 | 0.285 | 1.000 | 0.444 | 0.654 | 0.272 | 0.664 | 0.587 | 0.611 |
| Jordan | 3524 | 0.769 | 1.000 | 0.202 | 0.144 | 0.300 | 0.119 | 0.181 | 0.142 | 0.494 |
| Croatia | 3334 | 1.000 | 0.850 | 0.766 | 0.861 | 0.952 | 0.848 | 1.000 | 0.871 | 0.505 |
| Slovakia | 3129 | 0.794 | 0.108 | 0.397 | 0.319 | 0.411 | 0.331 | 0.328 | 0.402 | 0.429 |
| Luxembourg | 3028 | 1.000 | 0.483 | 0.550 | 0.867 | 0.639 | 0.552 | 0.626 | 0.175 | 0.433 |
| Cyprus | 2949 | 0.987 | 0.376 | 0.407 | 0.685 | 0.818 | 0.270 | 0.862 | 0.876 | 0.583 |
| Bulgaria | 2690 | 1.000 | 0.479 | 0.433 | 0.789 | 0.983 | 0.507 | 0.461 | 0.486 | 1.000 |
| Qatar | 2467 | 1.000 | 0.055 | 0.481 | 0.784 | 0.800 | 0.785 | 0.967 | 0.432 | 0.585 |
| Bangladesh | 2275 | 0.870 | 0.205 | 0.698 | 0.632 | 1.000 | 0.752 | 0.908 | 0.439 | 0.279 |
| Indonesia | 2266 | 0.590 | 0.350 | 0.439 | 0.776 | 1.000 | 0.233 | 0.765 | 0.527 | 0.466 |
| Lebanon | 2099 | 0.868 | 0.139 | 0.353 | 0.398 | 0.729 | 0.338 | 0.444 | 0.089 | 0.459 |

**Table A3.** *Cont.*

| Country | Count | Pattern | | | | | | | | |
|---|---|---|---|---|---|---|---|---|---|---|
| | | N/A | Fff | Mff | Fmm | Mmm | Ffm | Mfm | F | M |
| Macedonia | 2058 | 1.000 | 0.123 | 0.365 | 0.653 | 0.725 | 0.350 | 0.382 | 0.339 | 0.319 |
| Peru | 2049 | 0.724 | 0.283 | 0.399 | 0.678 | 0.437 | 0.436 | 0.444 | 1.000 | 0.386 |
| Ukraine | 1981 | 0.996 | 0.312 | 0.349 | 0.650 | 1.000 | 0.257 | 0.832 | 0.474 | 0.490 |
| Estonia | 1822 | 0.900 | 0.292 | 0.500 | 0.575 | 0.868 | 0.504 | 0.494 | 0.465 | 1.000 |
| Lithuania | 1768 | 0.789 | 0.069 | 0.376 | 0.553 | 0.764 | 0.140 | 0.681 | 0.618 | 0.877 |
| Kuwait | 1405 | 0.936 | 0.304 | 0.498 | 0.350 | 0.879 | 0.212 | 0.675 | 0.404 | 0.556 |
| Latvia | 1251 | 0.255 | 0.373 | 0.387 | 0.948 | 0.772 | 1.000 | 0.549 | 0.362 | 0.697 |
| Ecuador | 1190 | 1.000 | 0.034 | 0.444 | 1.000 | 0.724 | 0.498 | 0.645 | 0.808 | 0.740 |
| Philippines | 1046 | 0.720 | 0.176 | 0.181 | 0.257 | 0.638 | 0.553 | 0.301 | 0.716 | 1.000 |
| Niger | 1041 | 1.000 | 0.353 | 0.105 | 0.122 | 0.463 | 0.247 | 0.246 | 1.000 | 0.249 |
| Nigeria | 1032 | 1.000 | 0.353 | 0.108 | 0.122 | 0.470 | 0.247 | 0.246 | 1.000 | 0.251 |
| Mongolia | 968 | 0.694 | 0.147 | 0.534 | 0.773 | 0.897 | 0.568 | 0.807 | 0.091 | 1.000 |
| Iraq | 958 | 0.649 | 0.058 | 0.254 | 0.735 | 0.549 | 0.544 | 1.000 | 0.053 | 0.265 |
| Cuba | 943 | 0.268 | 0.334 | 0.466 | 0.369 | 0.766 | 0.550 | 0.550 | 0.000 | 1.000 |
| Venezuela | 936 | 0.366 | 0.121 | 0.059 | 0.062 | 0.116 | 0.188 | 0.106 | 0.078 | 0.059 |
| Uruguay | 887 | 0.648 | 0.273 | 0.362 | 0.515 | 1.000 | 0.493 | 0.455 | 0.227 | 0.857 |
| Iceland | 808 | 0.159 | 0.998 | 0.423 | 0.158 | 1.000 | 0.444 | 0.480 | 0.173 | 0.583 |
| Montenegro | 718 | 0.655 | 0.045 | 0.245 | 0.533 | 0.464 | 0.278 | 0.589 | 0.102 | 0.427 |
| Oman | 704 | 0.801 | 0.000 | 0.150 | 1.000 | 0.405 | 0.195 | 0.422 | 0.219 | 0.391 |
| Malta | 687 | 0.191 | 1.000 | 0.250 | 0.146 | 0.276 | 0.190 | 0.219 | 0.045 | 0.131 |
| Sri Lanka | 620 | 0.857 | 1.000 | 0.400 | 0.460 | 0.577 | 0.658 | 0.491 | 0.000 | 0.276 |
| Kazakhstan | 607 | 0.297 | 0.072 | 0.011 | 0.427 | 1.000 | 0.293 | 0.143 | 0.569 | 0.215 |
| Macau | 582 | 0.280 | 0.156 | 0.508 | 0.773 | 1.000 | 0.283 | 0.817 | 0.255 | 0.907 |
| Belarus | 572 | 0.427 | 0.056 | 0.097 | 0.139 | 0.069 | 1.000 | 0.094 | 0.094 | 0.032 |
| Puerto Rico | 483 | 0.662 | 0.307 | 0.501 | 0.781 | 0.725 | 0.082 | 0.720 | 0.000 | 0.391 |
| Saint Martin | 445 | 1.000 | 0.334 | 1.000 | 0.641 | 0.704 | 0.316 | 0.654 | 0.307 | 0.327 |
| Ethiopia | 380 | 0.524 | 0.000 | 0.135 | 0.210 | 0.638 | 0.125 | 0.337 | 1.000 | 0.054 |
| Small | 364 | 0.906 | 0.064 | 1.000 | 0.181 | 0.658 | 0.109 | 0.422 | 0.032 | 0.504 |
| Kenya | 324 | 0.179 | 0.067 | 0.417 | 0.015 | 0.523 | 0.346 | 1.000 | 0.000 | 0.240 |
| Armenia | 318 | 0.651 | 0.264 | 0.919 | 0.897 | 0.508 | 0.378 | 1.000 | 0.294 | 0.349 |
| Cameroon | 315 | 0.219 | 0.250 | 0.879 | 0.013 | 0.505 | 0.298 | 0.681 | 1.000 | 0.167 |
| Azerbaijan | 310 | 0.000 | 1.000 | 0.308 | 0.128 | 0.873 | 0.222 | 0.320 | 0.179 | 0.906 |
| Bosnia and Herzegovina | 302 | 0.000 | 0.004 | 1.000 | 0.661 | 0.532 | 0.462 | 0.077 | 0.036 | 0.407 |
| Palestine | 301 | 0.292 | 0.000 | 0.080 | 0.194 | 1.000 | 0.000 | 0.291 | 0.000 | 0.395 |
| Ghana | 299 | 0.361 | 0.154 | 0.327 | 0.040 | 0.063 | 1.000 | 0.072 | 0.013 | 0.024 |
| Costa Rica | 265 | 0.092 | 0.263 | 0.260 | 0.064 | 1.000 | 0.115 | 0.795 | 0.566 | 0.580 |
| Bahrain | 247 | 0.567 | 0.005 | 0.015 | 0.050 | 0.120 | 1.000 | 0.077 | 0.027 | 0.073 |
| Senegal | 194 | 1.000 | 0.000 | 1.000 | 0.119 | 0.271 | 0.643 | 0.384 | 0.000 | 0.010 |
| Brunei | 193 | 1.000 | 0.000 | 0.591 | 0.801 | 1.000 | 0.986 | 0.810 | 0.462 | 0.301 |
| Uganda | 187 | 1.000 | 0.126 | 0.042 | 0.211 | 0.240 | 0.208 | 0.398 | 0.055 | 0.552 |
| Myanmar | 187 | 0.989 | 0.253 | 0.108 | 0.058 | 0.038 | 0.243 | 0.494 | 0.000 | 1.000 |
| Mauritius | 184 | 1.000 | 0.144 | 0.042 | 0.187 | 0.725 | 0.267 | 0.140 | 0.926 | 1.000 |
| Libya | 171 | 1.000 | 0.000 | 0.932 | 0.302 | 0.364 | 0.000 | 0.393 | 0.000 | 0.217 |
| Fiji | 168 | 1.000 | 0.000 | 1.000 | 0.094 | 0.169 | 0.189 | 0.262 | 0.000 | 0.107 |
| Panama | 167 | 1.000 | 0.054 | 0.262 | 1.000 | 0.429 | 0.767 | 0.460 | 0.440 | 0.394 |
| Paraguay | 161 | 1.000 | 0.000 | 0.142 | 0.206 | 1.000 | 0.283 | 0.583 | 0.000 | 0.000 |
| Jamaica | 157 | 0.882 | 0.056 | 0.032 | 0.324 | 0.058 | 0.056 | 1.000 | 0.042 | 0.078 |
| Albania | 150 | 1.000 | 0.000 | 0.048 | 0.568 | 0.481 | 0.495 | 1.000 | 0.000 | 0.060 |
| Tanzania | 144 | 1.000 | 0.000 | 1.000 | 0.340 | 0.201 | 0.163 | 0.641 | 0.000 | 0.525 |
| Benin | 138 | 1.000 | 0.191 | 0.039 | 0.138 | 1.000 | 0.000 | 0.179 | 0.475 | 0.863 |
| Moldova | 134 | 0.581 | 0.124 | 0.094 | 0.000 | 0.394 | 0.194 | 0.293 | 0.000 | 1.000 |
| Liechtenstein | 125 | 1.000 | 0.000 | 0.743 | 0.425 | 1.000 | 0.008 | 0.622 | 0.000 | 0.292 |
| Yemale | 118 | 1.000 | 0.542 | 0.729 | 0.778 | 0.946 | 0.662 | 0.871 | 0.386 | 0.647 |
| Botswana | 117 | 1.000 | 0.620 | 0.710 | 0.759 | 0.993 | 0.641 | 0.832 | 0.613 | 0.709 |
| Sudan | 112 | 1.000 | 0.453 | 0.536 | 0.653 | 0.842 | 0.653 | 0.846 | 0.267 | 0.352 |
| Namibia | 111 | 1.000 | 0.428 | 0.887 | 0.676 | 0.902 | 0.637 | 0.741 | 0.252 | 1.000 |
| Syria | 105 | 1.000 | 0.550 | 0.652 | 0.617 | 0.766 | 0.507 | 0.772 | 0.618 | 0.642 |
| Trinidad and Tobago | 102 | 1.000 | 0.878 | 0.801 | 0.808 | 1.000 | 0.828 | 0.933 | 0.378 | 0.525 |

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
