# Peer review of "Gender-Related Differences in the Citation Impact of Scientific Publications and Improving the Authors’ Productivity"

_publications, doi:10.3390/publications11030037_

Round 1

Reviewer 1 Report

1.      The literature review in the introduction is well-written. However, I suggest the authors review and summarize more comprehensively the work on the relationship between gender compositions and citations. In particular, the authors may wish to consider citing the following paper (https://doi.org/10.1002/asi.24723), which investigates how a paper's first and last author gender pair affects its citations from Wikipedia.

2.      The math notations used in the paper could be made clearer and more comprehensible by using more internationally standard notation and tools like LaTeX to improve the layout.

3.      About gender imputation, the most common gender for a name may differ across countries. For example, Andrea is typically used for women in the U.S. and men in Italy. The authors may consider combining affiliation data with their gender imputation to improve accuracy or to discuss the potential impact of this limitation. Also, as a validation step, 100 names seem to be a small number and not persuasive. Are there any other studies reporting this service’s accuracy?

4.      The authors should consider moving section 3.1 to the methods and data section and removing duplicate or redundant descriptions of the data features.

5.      The distribution of publications appears to be heavily skewed towards computer science, which could be seen as a limitation. Computer vision, AI, etc should be subbranches of computer science, not at the same level.

6.      I did not fully understand the purpose of using normalized Shannon entropy. It is expected that the distribution of gender compositions is dispersed. To measure the representativeness, the authors may need to obtain the overall population’s distribution and then compare.

7.      It would be helpful for the authors to provide a more detailed explanation of how they aggregate PageRank scores at the country level, as in Table 4-5. Descriptive statistics for the PageRank scores may also be necessary as there may be highly-cited outliers.

8.      With regard to pairwise comparison, the authors should consider comparing additional pairs such as Fmm/Mmm, Fff/Mff, Mmm/Fmm, Mff/Fff, etc. Also, it is expected to provide statistical test results to determine if the differences are significant.

9.      For Figure 4, I suggest the authors split the period into shorter intervals and demonstrate the trend in lower granularity.

10.   The authors state in discussion that women face disadvantages in science due to childbearing and childcaring. I agree with this point, but there are no external references to support it. I suggest referencing this paper (https://doi.org/10.7554/eLife.78909) which provides further empirical evidence and discussion of this issue.

11.   Please check the reference list. Some references lack necessary elements.

It reads OK overall, but some sentences need further checking and revising.

Author Response

1. The article has been reviewed. Reference is made in section 4.1.

2. The formulas were reviewed. They were typed in MathType, which meets the requirements of the editor. In case of problems with the display of formulas, the authors will work with the editors of the journal.

3. The number 100 is explained by the fact that the Gender API allows you to check 100 names per month for free. As part of the study, the main name gendering tool was genderize.io, which is free. Regarding the same names in different countries, the discussion was indicated in section 4.2.

4. Part of section 3.1 has been moved to a new section 2.3. Added section 3.1.
5. Subject area in this dataset was already defined by the authors of the study Tang, J., Zhang, J., Yao, L., Li, J., Zhang, L., & Su, Z. (2008). ArnetMiner : Extraction and Mining of Academic Social Networks. Proc. Fourteenth ACM SIGKDD International Conference on Knowledge Discovery and Data Mining (SIGKDD'2008), 990 – 998. Some of the indicated subject areas may be part of other, more general subject areas. For example, Artificial intelligence can be a subfield of Computer science. This was indicated in section 3.1.

6. The dataset was examined to fulfill the various requirements within the proposed subsets defined by the defined patterns. For this, the normalized Shannon entropy was calculated. Measure the representativeness with the overall population's distribution out of scope of this research.

7. Statistical characteristics were calculated and described in section 3.2. and in the discussion.

8. Two more ratios between two patterns were calculated. They were added to Table 6.

9. We divided the period into 2 intervals in Figure 4.

10. The link has been added

Reviewer 2 Report

The work is very interesting, the idea is well-developed and well-conducted in the study.

A few things to explain/correct:

1.  Pg. 4 first paragraph: "[…] following research stages were implemented: 1. Calculate the citation impact for each scientific publication in the citation network. For this, a method based on calculating the number of citations of scientific publications was used...]. What method are you referring to? It can be just a phrasing problem in English, but otherwise, the authors should specify.

2. Table 3 (but also tables 4, 5, and 6) - are they incomplete, in the sense of evading part of the data? If so, this should be marked in column 2, by mentioning the data discontinuity or by specifying the type of sampling.

3. In all 3 Appendices (A2, A2, A3), there is a country called "Armaleia" that appears in the list of countries. This does not exist, obviously, so could not have been part of the database. It seems like a typo, but it would also be interesting to see the raw data of the study, to eliminate any suspicion.

In all 3 Appendices (A2, A2, A3), there is a country called "Armaleia" that appears in the list of countries. This does not exist, obviously, so could not have been part of the database. It seems like a typo, but it would also be interesting to see the raw data of the study, to eliminate any suspicion.

No publication before checking this out.

Author Response

1. Chapter 1 was clarified.

2. Tables 3, 4, 5 show part of the data. The full versions of the tables are listed in the appendices. This is done to improve the perception of the article for the reader.

3. Armaleia was a typo

Round 2

Reviewer 1 Report

No further comments.

No further comments.

Reviewer 2 Report

Improvement is obvious.

Good luck!